# A-Bench: Are LMMs Masters at Evaluating AI-generated Images?

**Zicheng Zhang**[1*]**, Haoning Wu**[2*]**, Chunyi Li**[1]**, Yingjie Zhou**[1]**, Wei Sun**[1]**,**
**Xiongkuo Min**[1]**, Zijian Chen**[1]**, Xiaohong Liu**[1]**, Weisi Lin**[2]**, Guangtao Zhai**[1†]**,**
[1]Shanghai Jiaotong University, [2]Nanyang Technological University
[*]Equal contribution. [†]Corresponding authors. Project Page: *https://github.com/Q-Future/A-Bench*.

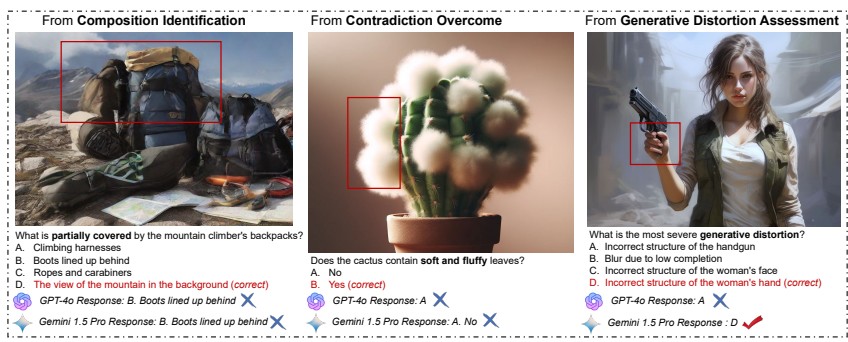

Figure 1: Error cases from the **A-Bench**.

## Abstract

How to accurately and efficiently assess AI-generated images (AIGIs) remains a critical challenge for generative models. Given the high costs and extensive time commitments of user studies, many researchers have turned towards employing large multi-modal models (LMMs) as AIGI evaluators, the precision and validity of which are still questionable. Furthermore, traditional benchmarks often utilize mostly natural-captured content rather than AIGIs to test the abilities of LMMs, leading to a noticeable gap for AIGIs. Therefore, we introduce **A-Bench** in this paper, a benchmark designed to diagnose *whether LMMs are masters at evaluating AIGIs*. Specifically, **A-Bench** is organized under two key principles: 1) Emphasizing both high-level semantic understanding and low-level visual quality perception to address the intricate demands of AIGIs. 2) Various generative models are utilized for AIGI creation, and various LMMs are employed for evaluation, which ensures a comprehensive validation scope. Ultimately, 2,864 AIGIs from 16 text-to-image models are sampled, each paired with question-answers annotated by human experts. We hope that **A-Bench** will significantly enhance the evaluation process and promote the generation quality for AIGIs.

## 1 Introduction

*One look is worth a thousand words.* Inspired by this age-old adage, numerous researchers dedicate their efforts to developing text-to-image (T2I) models that vividly bring text to life through imagery. These T2I models, driven by free-form text prompts, aim to create images that **accurately align with the text and showcase high perceptual quality.** Innovations such as AlignDRAW (Mansimov et al., 2015) and the text-conditional GAN (Reed et al., 2016) have introduced differential architecture for image generation. The field continues to advance with the development of stable diffusion models (Saharia et al., 2022; Rombach et al., 2022b), significantly propelling T2I technology forward. On the commercial front, major corporations leverage vast-scale data to launch stunningly effective T2I models, such as DALL-E (Ramesh et al., 2022), Midjourney (Holz, 2023), Parti (Yu et al., 2022), etc. However, despite their diversity and widespread adoption, all these advanced T2I models occasionally *face issues of low alignment with prompts and low perceptual quality* in creating AI-generated images (AIGIs), necessitating careful evaluation and improvement.

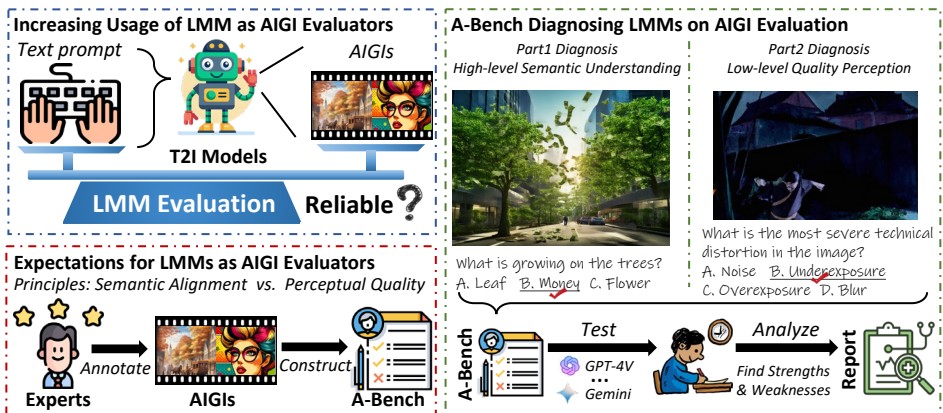

Figure 2: The proposed **A-Bench** is designed to find out whether LMMs are reliable for T2I AIGI evaluation. Instead of directly assessing the performance of LMM-based metrics, we evaluate the **LMMs themselves behind** by examining whether the *fundamental questions regarding semantic understanding and quality perception can be correctly answered.* Based on the benchmark results, we can then 'diagnose' the strengths and weaknesses across various LMMs.

The alignment and quality evaluation of AIGIs present significant challenges that *small expert models* attempt to address. Although these *small expert models* offer some solutions, they possess inherent drawbacks and often fail to meet contemporary demands. Specifically, for alignment assessment, CLIP-based similarity models struggle with accurately judging alignment, particularly with complex text prompts (Radford et al., 2021b). When it comes to quality evaluation, traditional image quality/aesthetic assessment methods (IQA/IAA) are not capable of identifying AIGI-generative distortions (Wu et al., 2023a;b), rendering them unsuitable for this specialized task.

Many researchers are increasingly relying on large language models (LLMs) and large multi-modal models (LMMs) for their human-like processing capabilities, which are presumed to enable accurate judgments of alignment and quality in generated content. Consequently, many LMM-based evaluators have been developed, including VIE-Score (Ku et al., 2023), Prometheus (Kim et al., 2023), VQAScore (Lin et al., 2024), GPT4V-Eval (Zhang et al., 2023b), TIFA (Hu et al., 2023), and Davidsonian Graph (Cho et al., 2023), etc. However, a fundamental question remains:

*Are LMMs reliable for evaluating T2I AIGIs?*

These LMM-based metrics traditionally employ evaluation criteria such as SRCC/PLCC to determine their reliability. However, this approach only reveals *how well the metrics perform, without shedding light on their specific strengths and weaknesses.* To address this gap, we propose conducting a comprehensive *'diagnostic'* benchmark → **A-Bench**, focusing on LMMs' capabilities in semantic understanding and quality assessment. Rather than directly evaluating these LMM-based metrics, we focus on *the LMMs themselves behind.* We move away from computing SRCC/PLCC criteria and instead examine *whether the fundamental perceptual questions can be correctly answered*, which is the core basis of all LMM-based evaluators. To initiate our exploration on the AIGI evaluation abilities of LMMs, we first construct the **A-Bench** centered on a pivotal question:

*What do we expect from LMMs as AIGI evaluators?*

The answer lies in the capabilities of **semantic alignment and quality evaluation**. We then define two key diagnostic subsets: **A-Bench**$^{P1}$→*high-level semantic understanding*, and **A-Bench**$^{P2}$→*low-level quality perception.* For *high-level semantic understanding*, **A-Bench**$^{P1}$ targets three critical areas: *Basic Recognition*, *Bag-of-Words Pitfalls Discrimination*, and *Outside Knowledge Realization*, which are designed to progressively test the LMM's capability in AIGI semantic understanding, moving from simple to complex prompt-related content. For *low-level quality perception*, **A-Bench**$^{P2}$ concentrates on *Technical Quality Perception*, *Aesthetic Quality Evaluation*, and *Generative Distortion Assessment*, which are designed to cover the common and AIGI-specific quality problems. The aspect selection is meticulously designed to encompass the most prevalent application scenarios. Specifically, a comprehensive dataset of 2,864 AIGIs sourced from various T2I models is compiled, including 1,408 AIGIs for **A-Bench**$^{P1}$ and 1,456 for **A-Bench**$^{P2}$. Each AIGI is

paired with a question-answer set annotated by human experts. We then test 23 prominent LMMs, including both *open-source* and *closed-source* models, on the **A-Bench**. From the results that the best LMM still falls behind humans by a large margin, we can derive the following conclusion:

*LMMs are still not masters at evaluating AIGIs.*

All LMMs lag behind even the poorest human performance on **A-Bench**, and there is a substantial disparity between *open-source* LMMs and *closed-source* LMMs. The performance across different subcategories fluctuates for both **A-Bench**$^{P1}$ and **A-Bench**$^{P2}$, indicating that LMMs are not yet robust for different evaluation scenarios for AIGIs. There remains a considerable gap and significant room for improvement before LMMs can be considered masters of evaluating AIGIs.

In summary, we systematically explore the capabilities of LMMs in semantic understanding and quality perception, both crucial for their role as AIGI evaluators. These two essential capabilities constitute the core of the proposed **A-Bench**, the first 'diagnostic' benchmark specifically designed for LMM assessment in AIGI evaluation. Our contributions are summarized as follows:

- We carry out the **first 'diagnostic' benchmark** on AIGI evaluation for LMMs, which consists of 2,864 AIGIs (sampled from various T2I models) paired with question-answer sets on both high-level semantic understanding and low-level quality perception.

- A detailed discussion is made about **what to 'diagnose'**. Semantic understanding is subdivided into *Basic Recognition, Bag-of-Words Pitfalls Discrimination*, and *Outside Knowledge Realization* while quality perception is broken down into *Technical Quality Perception, Aesthetic Quality Evaluation*, and *Generative Distortion Assessment*.

- From the benchmark results, several insights are gleaned, which can enable us to diagnose various issues with different LMMs and assist in their improvement for AIGI evaluation.

## 2 RELATED WORKS

### 2.1 LARGE MULI-MODAL MODELS

Large language models (LLMs), such as GPT-4 (OpenAI, 2023), T5 (Chung et al., 2022), and LLaMA (Touvron et al., 2023), exhibit exceptional linguistic capabilities in general human knowledge domains. By integrating visual input via CLIP (Radford et al., 2021a) and additional adaptation modules, large multi-modal models (LMMs) (Li et al., 2023a; Gao et al., 2023; Liu et al., 2023b; Dai et al., 2023; Zhang et al., 2023a) are capable of addressing diverse multi-modal tasks, including image captioning, visual question answering, visual segmentation, visual classification, visual reasoning, etc. Namely, OpenFlamingo (Awadalla et al., 2023) initially integrates several gated cross-attention dense blocks into the pretrained language encoder layers. InstructBLIP (Dai et al., 2023) extends BLIP-2 (Li et al., 2023c) by incorporating vision-language instruction tuning. To further develop *open-source* LMMs, many works have employed GPT-4 (OpenAI, 2023) to create data for vision-language tuning, such as LLaVA series (Liu et al., 2023b;a; 2024). However, whether these LMMs are masters at evaluating T2I AIGIs is still questionable, which needs further investigation.

### 2.2 MULTI-MODAL BENCHMARKS

Benchmarks such as COCO Caption (Chen et al., 2015) and Nocaps (Agrawal et al., 2019) evaluate the capability of models to generate textual descriptions for images. Subsequently, benchmarks like GQA (Hudson & Manning, 2019) and OK-VQA (Marino et al., 2019) focus on visual question answering, assessing multi-modal models' visual perception and reasoning abilities. Further complexities are added in benchmarks such as TextVQA (Singh et al., 2019) and ScienceQA (Lu et al., 2022), which incorporate OCR tasks and commonsense reasoning, respectively. MME (Fu et al., 2023) and MMbench (Liu et al., 2023c) provide comprehensive evaluations of LMMs across various subtasks. Additionally, MMMU (Yue et al., 2023) targets extensive multi-disciplinary tasks that require college-level knowledge and sophisticated reasoning. More recently, Q-Bench (Wu et al., 2023a) focuses specifically on assessing the low-level visual perception capabilities of LMMs. Despite these efforts, there is still a gap in systematic benchmarks for assessing the abilities of LMMs in AIGI evaluation, prompting the development of **A-Bench** to address this shortfall.

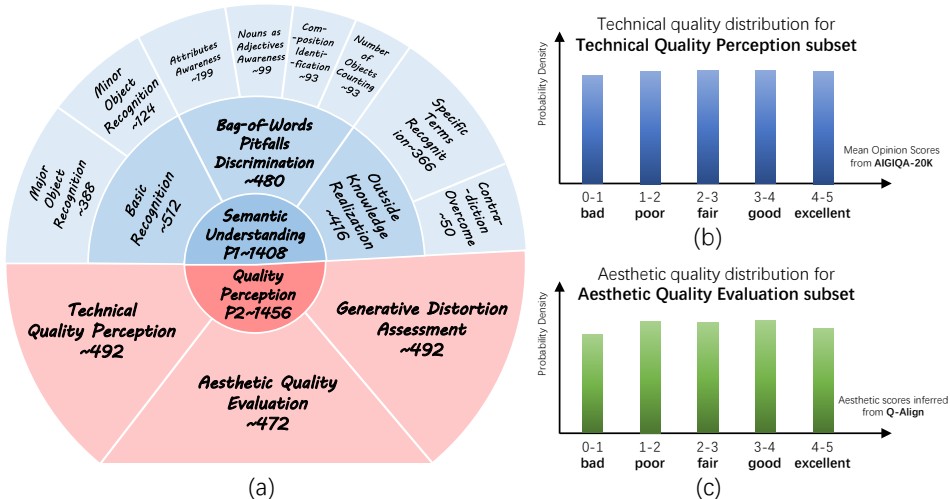

Figure 3: Illustration of focused aspects and corresponding quality distributions for **A-Bench**. The focused aspects and the amount of AIGIs employed are shown in (a). The quality scores of AIGIs sampled for **Technical Quality Perception** and **Aesthetic Quality Evaluation** subsets are obtained from AIGIQA-20K (Li et al., 2024) and predicted from Q-Align (Wu et al., 2023c) respectively.

# 3 CONSTRUCTING THE A-BENCH

## 3.1 KEY PRINCIPLES

**Covering High-level and Low-level Attributes.** The demand for generating images has become increasingly stringent, with requirements for *not only accurate adherence to prompt specifications but also high visual quality of AIGIs*. To ascertain whether LMMs can effectively evaluate whether AIGIs meet these criteria, it is essential to assess their capabilities in both **high-level semantic understanding** and **low-level quality perception**. High-level semantic understanding encompasses basic recognition and the integration of external knowledge, whereas low-level quality perception involves the identification of technical quality, aesthetic appeal, and generative distortions. The detailed focused aspects can be overviewed in Fig. 3 (a).

**Ensuring Diverse AIGI Scope.** Considering the variety of current generative models and their application scenarios, we have selected a broad range of mainstream text-to-image (T2I) generation models to produce AI-generated images (AIGIs). To assess high-level semantic understanding, we design prompts rich in content to ensure diversity among the generated images. For evaluating low-level quality perception, we employ uniform sampling to encompass a wide spectrum of visual quality and the corresponding quality distributions are illustrated in Fig 3 (b) and (c). Throughout the benchmarking process, we test multiple *open-source* and *closed-source* LMMs to guarantee a comprehensive evaluation. These measures ensure that our proposed **A-Bench** encompasses a diverse and extensive scope. More details about AIGIs collection can be referred to in Sec. A.1.

## 3.2 FOCUSED ASPECTS

The key evaluation aspects of T2I models involve image-text alignment and image visual quality, which correspond to high-level semantic understanding and low-level quality perception abilities. Some representative examples regarding the subcategories discussed below are exhibited in Fig. 4.

### 3.2.1 HIGH-LEVEL SEMANTIC UNDERSTANDING

To evaluate whether LMMs can effectively assess image-text alignment, we implement the **A-Bench**[P1], which consists of 1,408 challenging multi-modal question-answer pairs that focus on high-level semantic understanding for AIGIs. The high-level semantic understanding can be divided into the following subcategories, moving from simple to complex prompt-related content:

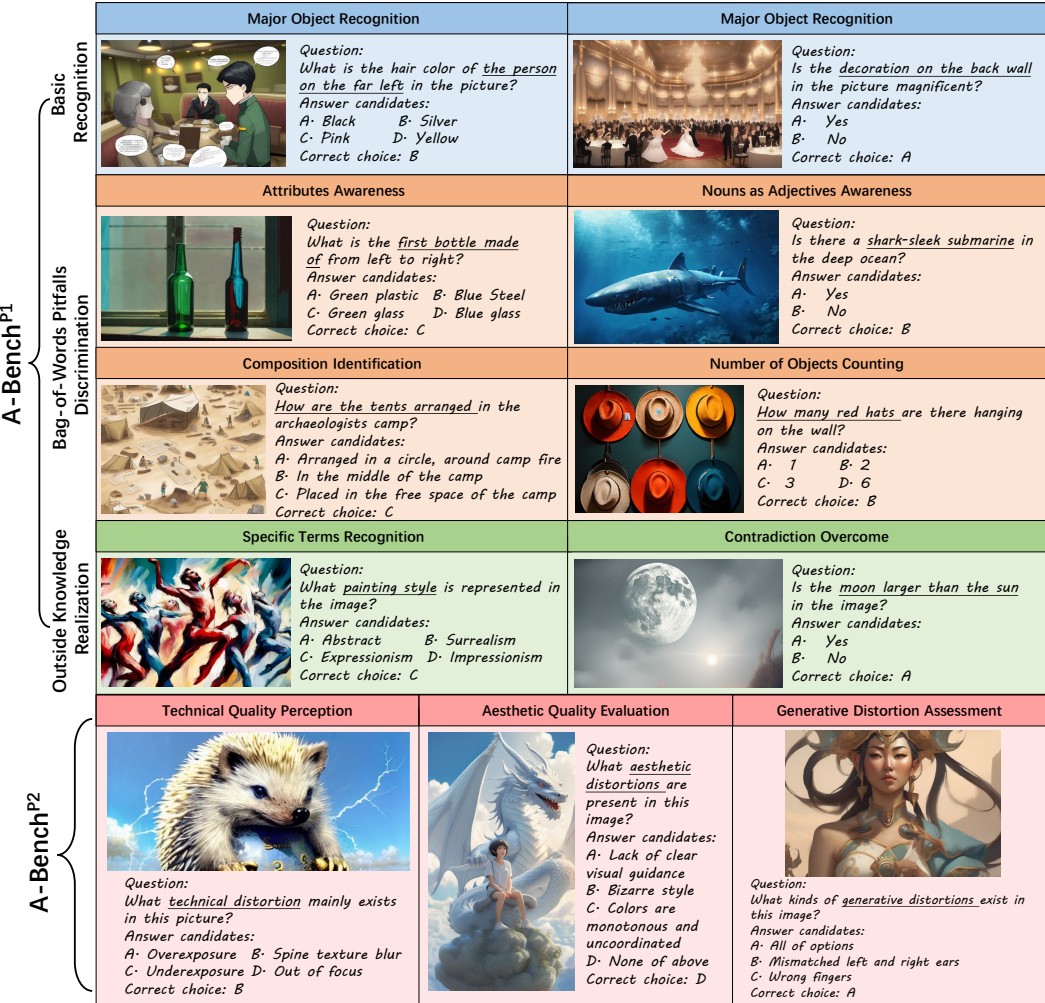

Figure 4: Examples of **A-Bench**. Each AIGI is accompanied by a question-answer pair.

**Basic Recognition.** This aspect concentrates on the fundamental semantic understanding of AIGIs (Nichol et al., 2021; Saharia et al., 2022), which can be subdivided into two distinct areas based on the objects of interest: 1) **Major Object Recognition**, which involves *recognizing the primary objects in the image*, such as humans or objects depicted in the foreground. 2) **Minor Object Recognition**, which *pertains to the identification of less-prominent objects within the image*, such as background elements or secondary characters.

**Bag-of-Words Pitfalls Discrimination.** This dimension focuses on the discriminative semantic understanding of AIGIs crafted with *Bag-of-words prompts* (encompassing rich descriptive attributes or complex object relationships (Qu et al., 2024)). This can be subdivided into the following aspects related to the crucial points of T2I generation alignment: 1) **Attributes Awareness**, defined as *the capability to accurately identify the attributes of objects in AIGIs* (Xu et al., 2023; Liu et al., 2023c). 2) Additionally, given that T2I models may incorrectly interpret nouns as adjectives, resulting in *the unwanted generation of objects instead of the intended attributes* (Chatterjee et al., 2025; Motamed et al., 2023), we have also introduced a dimension called **Nouns as Adjectives Awareness** to address this issue. 3) **Composition Identification**, recognized as the ability to *correctly comprehend the compositional relationships* such as orientation, occlusion, size comparison, and spatial arrangement (Wang et al., 2024b; Zhang et al., 2024). 4) **Number of Objects Counting**, regarded as *the ability to accurately count the specified objects in the image*, which is crucial for assessing whether the AIGI aligns with the numerical specifications of the prompt (Binyamin et al., 2024).

**Outside Knowledge Realization.** This aspect emphasizes the reasoning ability to utilize external knowledge not directly depicted in the images (Schwenk et al., 2022), and can be broken down into the following dimensions: 1) **Specific Terms Recognition**: This involves *identifying specific scenes and objects* related to distinct domains such as geography, sports, science, materials, food, everyday life, creatures, brands, and styles. 2) **Contradiction Overcome**, recognized as the *ability to correctly interpret AIGIs even when their content contradicts established world knowledge*, which is particularly crucial for evaluating AIGIs generated from controversial prompts (Hou et al., 2024).

### 3.2.2 LOW-LEVEL QUALITY PERCEPTION

Conversely, to determine the ability of LMMs on image visual quality, we conduct the **A-Bench**$^{P2}$, comprising 1,456 challenging multi-modal question-answer pairs centered on low-level quality perception for AIGIs, which can be categorized into the following aspects: 1) **Technical Quality Perception** This indicates the *low-level characteristics that directly degrade the quality of images*, such as blur, noise, exposure, etc (Su et al., 2021; Ying et al., 2020). 2) **Aesthetic Quality Evaluation** This indicates the *attributes that affect the aesthetic appeal of AIGIs and evoke varied human feelings*, which include color, lighting, etc (Huang et al., 2024). 3) **Generative Distortion Assessment** This indicates the *unexpected AIGI-specific distortions* (Chen et al., 2023b; Li et al., 2023b; 2024), such as generative blur caused by low completion, confusing geometry structure, unnaturalness, etc.

### 3.3 QUESTION COLLECTION

**Question Type** In the **A-Bench**, two types of question formats are utilized, including *Yes-or-No* questions and *What* questions. The *Yes-or-No* questions (accounting for 25.9%) are used to evaluate the fundamental judgment abilities of LMMs while the *What* questions (accounting for 74.1%) are more complicated and require LMMs to gain a more comprehensive understanding of the AIGIs.

**Human Expert Annotation** We have assembled a team of 15 human annotators, each with expert experience in AIGI evaluation, to develop questions for **A-Bench**. This annotation process is conducted in a controlled laboratory environment, ensuring consistency and reliability. Annotators are tasked with designing questions specific to the sub-categories of the AIGIs under review, utilizing their extensive knowledge to determine the content and format of each question. To ensure the highest quality and suitability, each question undergoes a rigorous review process, with at least three other expert annotators double-checking it. More details can be acquired in Sec. A.3.

**Question Response** Specifically, the example input query to LMMs can be exemplified as:

*#User: What painting style is represented in the image? $|IMAGE\_TOKEN|$*
*A. Abstract    B. Surrealism    C. Expressionism    D. Impressionism*
*Answer with the option's letter from the given choices directly.*

The answer candidates and correct answers are shuffled during the evaluation process. Since the responses from LMMs can be in various forms (if the correct choice is C) such as '*C*', '*Expressionism*', '*The painting style of image is expressionism*', etc., we employ a GPT-assisted choice evaluation technique proposed in (Liu et al., 2023c; Wu et al., 2023a) to validate the correctness of LMMs responses. More details are shown in Sec. A.4.

## 4 EXPERIMENT RESULTS

### 4.1 BENCHMARK CANDIDATES

To ensure the results are comprehensive and up-to-date, we select the widely used LMMs for benchmarking. The **Proprietary LMMs** (*closed-source*) include Gemini 1.5 Pro (Reid et al., 2024), GPT-4v (OpenAI, 2023), GPT-4o (2024-05-13) (OpenAI, 2024), and Qwen-VL-Max (Bai et al., 2023). The **Open-source LMMs** include Qwen2-VL-72B (*Qwen2-72B*) (Wang et al., 2024a), MiniCPM-V2.6 (*Qwen2-7B*) (Yao et al., 2024), InternVL2-40B (*Nous-Hermes-2-Yi-34B*) (Chen et al., 2024), Ovis1.5 (*Llama3-8B*), LLaVA-OneVision (*Qwen2-7B*) (Lu et al., 2024b), CogVLM2-19B (*Llama3-8B*) (Wang et al., 2023), IDEFICS-2 (*Mistral-7B-Instruct-v0.2*) (Huggingface, 2023), DeepSeek-VL-7B (Lu et al., 2024a), InternLM-XComposer2-VL (Dong et al., 2024), LLaVA-NeXT (*Llama3-*

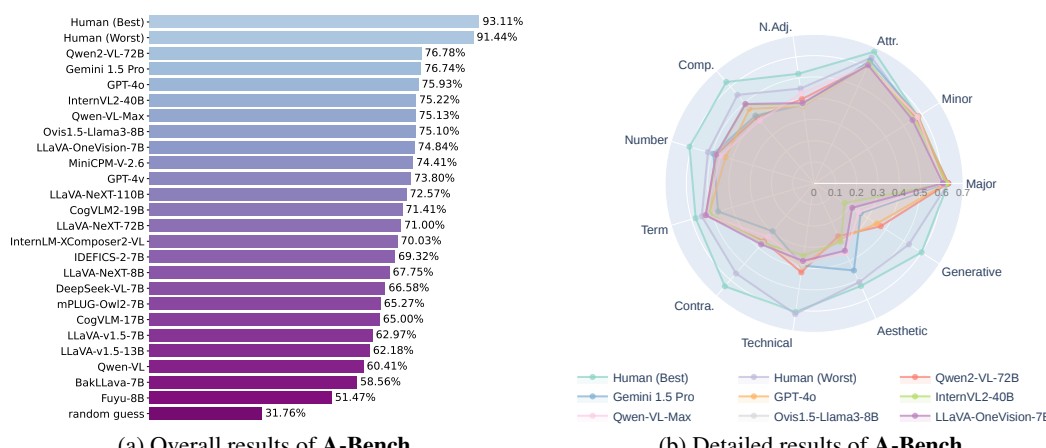

(a) Overall results of **A-Bench**.    (b) Detailed results of **A-Bench**.

Figure 5: *A Quick Look* of the **A-Bench** outcomes. (a) showcases a comparative analysis of the overall accuracy between humans, 23 selected LMMs (both *closed-source* and *open-source*), and *random guess*. (b) displays a radar chart that details the accuracy performance (**subtracting the accuracy** of *random guess*) of the **top-7** LMMs across various subcategories of **A-Bench**.

*8B*), LLaVA-NeXT (*Qwen-72B*), LLaVA-NeXT (*Qwen-110B*) (Liu et al., 2024), mPLUG-Owl2 *(LLaMA-7B)* (Ye et al., 2023), LLaVA-v1.5 (*Vicuna-v1.5-7B*), LLaVA-v1.5 (*Vicuna-v1.5-13B*) (Liu et al., 2023a), CogVLM-17B (*Vicuna-v1.5-7B*) (Wang et al., 2023), Qwen-VL *(Qwen-7B)* (Bai et al., 2023), BakLLava (*Mistral-7B*) (Liu et al., 2023b), and Fuyu-8B (*Persimmon-8B*) (Adept, 2023). All LMMs are tested with zero-shot setting. It's worth noting that the instruction prompt might slightly differ for different LMMs according to the official setting.

## 4.2 HUMAN PERFORMANCE

For human performance on **A-Bench**, we conduct a user-study experiment with five ordinary people in a controlled laboratory setting. Initially, participants familiarize themselves with the tasks through exposure to similar cases. Subsequently, they select the appropriate responses for the questions posed in the **A-Bench**. To maintain consistency with the conditions experienced by LMMs, the order of questions is randomized, and participants receive no additional information beyond the AIGIs, questions, and answer options. The *best* and *worst* performance is included for comparison. More details about acquiring human performance can be referred to in Sec.A.5.

## 4.3 FINDINGS OF A-BENCH

**General Observation: Proprietary LMMs deliver performance comparable to the best open-source LMMs.** A concise overview of the **A-Bench** results is provided in Fig. 5, revealing several general insights: 1) All LMMs significantly outperform the *random guess*, indicating their capabilities in handling AIGI evaluation, with Qwen2-VL-72B leading, closely followed by Gemini 1.5 Pro and GPT-4o. Notably, among the *open-source* LMMs, which are preferred for AIGI evaluations due to their accessibility and modifiability, Qwen2-VL-72B stands out, even outperforming the best *closed-source* competitors. 2) Even the lowest performance by humans surpasses that of all LMMs, with a noticeable 14.62% gap compared to the top-performing LMM, Qwen2-VL-72B, indicating that LMMs are still far from adequately performing AIGI evaluation as humans. 3) A closer examination of the radar chart in Fig. 5 (b) shows that top LMMs exhibit varied performances across different sub-categories, suggesting a lack of robustness, while humans show more consistent and balanced performance across these categories, highlighting areas where LMMs need further improvement.

**Findings of A-Bench[P1]: LMMs excel at basic recognition tasks but tend to be less effective when it comes to nuanced semantic understanding.** The performance results of LMMs on the **A-Bench**[P1] subset, as detailed in Table 1, reveal several key insights: 1) Almost all LMMs show

Table 1: Benchmark results on the **A-Bench**$^{P1}$ subset, which reveal the high-level semantic understanding abilities across LMMs. The best performance is marked in **bold** and the second performance is underlined for both proprietary and open-source LMMs respectively.

| Categories | Basic Recognition | | Bag-of-Words | | | | Outside Knowledge | | Overall↑ |
|---|---|---|---|---|---|---|---|---|---|
| LMM (LLM) | Major↑ | Minor↑ | Attr.↑ | N. Adj.↑ | Comp.↑ | Number↑ | Term↑ | Contra.↑ | |
| HUMAN (WORST) | 95.18% | 94.24% | 96.78% | 88.70% | 85.49% | 82.46% | 81.76% | 88.91% | 92.40% |
| HUMAN (BEST) | 95.40% | 95.21% | 99.42% | 95.17% | 93.34% | 91.73% | 84.29% | 96.05% | 94.02% |
| **Proprietary LMMs:** | | | | | | | | | |
| GEMINI 1.5 PRO | 93.82% | 95.18% | **94.35%** | 80.27% | 72.14% | **79.35%** | 72.88% | 61.56% | 84.70% |
| GPT-4v | 92.95% | **96.00%** | 87.40% | 82.67% | 64.39% | 68.84% | 77.60% | 66.73% | 83.60% |
| GPT-4o (2024-05-13) | **94.34%** | 95.14% | 91.99% | 79.54% | **76.40%** | 73.30% | 77.47% | **68.59%** | **85.44%** |
| QWEN-VL-MAX | 92.56% | 94.75% | 91.99% | **85.78%** | 68.94% | 75.85% | **78.94%** | 65.05% | 84.47% |
| **Open-source LMMs:** | | | | | | | | | |
| Qwen2-VL-72B (Qwen2-72B) | **95.15%** | 94.61% | 92.31% | 83.66% | 71.37% | 78.20% | 79.12% | 68.99% | **86.02%** |
| MiniCPM-V2.6 (Qwen2-7B) | 93.01% | 93.22% | 93.44% | 81.21% | 78.31% | 77.06% | **79.32%** | 67.86% | 84.98% |
| InternVL2-40B (Nous-Hermes-2-Yi-34B) | 94.86% | 93.87% | **93.56%** | 80.32% | **79.88%** | 78.01% | 77.44% | 69.54% | 85.17% |
| Ovis1.5 (Llama3-8B) | 92.79% | 92.26% | 92.12% | 80.55% | 78.61% | **78.59%** | 78.34% | 69.87% | 85.08% |
| LLaVA-OneVision (Qwen2-7B) | 92.53% | 92.01% | 92.07% | 81.12% | 79.33% | 77.98% | 79.02% | **69.91%** | 84.88% |
| CogVLM2-19B (Llama3-8B) | 93.31% | 92.70% | 89.97% | 75.41% | 64.63% | 66.63% | 75.88% | 61.54% | 82.55% |
| IDEFICS-2 (Mistral-7B-Instruct-v0.2) | 89.92% | 91.87% | 86.50% | 81.36% | 61.36% | 71.04% | 73.31% | 62.91% | 80.14% |
| DeepSeek-VL-7B | 91.48% | 91.15% | 82.44% | 83.73% | 63.38% | 69.91% | 75.40% | 60.32% | 81.42% |
| InternLM-XComposer2-VL (InternLM2) | 92.79% | **95.21%** | 86.38% | 82.64% | 68.87% | 72.22% | 70.77% | 64.35% | 81.89% |
| LLaVA-NeXT (Llama3-8B) | 92.72% | 92.40% | 91.15% | 83.62% | 61.04% | 67.07% | 76.23% | 62.94% | 82.88% |
| LLaVA-NeXT (Qwen-72B) | 94.37% | 92.72% | 91.49% | 81.61% | 62.40% | 73.39% | 77.15% | 61.44% | 83.99% |
| LLaVA-NeXT (Qwen-110B) | 93.83% | 91.10% | 90.43% | **84.71%** | 67.76% | 67.70% | 76.25% | 64.28% | 83.66% |
| mPLUG-Owl2 (LLaMA-7B) | 85.29% | 86.26% | 83.87% | 79.66% | 53.73% | 57.85% | 71.14% | 58.47% | 76.40% |
| LLaVA-v1.5 (Vicuna-v1.5-7B) | 87.82% | 88.65% | 83.86% | 75.41% | 61.39% | 65.67% | 74.76% | 62.69% | 78.86% |
| LLaVA-v1.5 (Vicuna-v1.5-13B) | 88.60% | 89.57% | 86.48% | 79.52% | 62.33% | 58.82% | 74.81% | 61.56% | 79.72% |
| CogVLM-17B (Vicuna-v1.5-7B) | 90.38% | 95.17% | 85.89% | 77.47% | 49.56% | 47.82% | 73.34% | 61.34% | 78.61% |
| Qwen-VL (Qwen-7B) | 86.14% | 86.32% | 81.38% | 77.47% | 52.72% | 61.22% | 71.61% | 57.32% | 76.39% |
| BakLLava (Mistral-7B) | 88.91% | 81.31% | 77.42% | 73.81% | 52.18% | 62.32% | 68.37% | 49.02% | 74.33% |
| Fuyu-8B (Persimmon-8B) | 81.41% | 68.27% | 66.72% | 57.45% | 42.24% | 48.32% | 61.16% | 29.65% | 63.12% |
| *random guess* | 32.27% | 37.22% | 31.03% | 42.82% | 29.85% | 29.78% | 26.51% | 32.13% | 30.80% |

good performance in *Basic Recognition*, suggesting that they are quite adept at fundamental semantic understanding, which includes recognizing foreground and background objects in AIGIs. 2) However, their effectiveness diminishes in more complex tasks such as *Bag-of-Words*, particularly in subcategories like *Nouns as Adjectives Awareness*, *Composition Identification*, and *Number of Objects Counting*. These areas require deeper semantic understanding and reasoning, which is critical as users often employ complex prompts that include such nuanced elements. The LMMs' underperformance here indicates potential challenges in accurately aligning AIGIs with user prompts. 3) Additionally, *Outside Knowledge* poses significant challenges, with LMMs generally achieving unsatisfactory performance in the *Contradiction Overcome* subcategory, where AIGIs contain content that defies common sense, requiring LMMs to override their prior knowledge to respond correctly. The subcategory *Specific Terms* tests the knowledge base of LMMs, where proprietary LMMs generally perform better due to being trained on more recent and extensive datasets. 4) Therefore, to improve the evaluation of text alignment in AIGIs using LMM, it is recommended to simplify overly complex prompts. By employing a divide-and-conquer approach to break down complex prompts into shorter ones, sequential judgment can effectively enhance accuracy.

**Findings of A-Bench**$^{P2}$**: LMMs are poor quality evaluators.** The performance results of LMMs on the **A-Bench**$^{P2}$ subset, as shown in Table 2, illustrate a notable disparity in capabilities: 1) There is a significant performance gap of approximately 23.10% between the top-performing LMMs and human evaluators, highlighting that LMMs lag considerably in quality perception and struggle to accurately assess the quality of AIGIs. 2) Furthermore, most LMMs exhibit their weakest performance in the *Generative Distortion Assessment* subcategory (except Qwen2-VL-72B), suggesting their ineffectiveness at identifying unexpected generative distortions, such as unnatural appearances and incorrect geometric structures. 3) Interestingly, while humans generally perform better in *Technical Quality Perception* compared to *Aesthetic Quality Evaluation*, LMMs show similar performance levels in both subcategories (except Qwen2-VL-72B and MiniCPM-V2.6). This difference likely stems from the more objective nature of technical quality assessments, which leads to more con-

Table 2: Benchmark results on the **A-Bench**$^{P2}$ subset, which reflect the low-level quality perception abilities across LMMs. The best performance is marked in **bold** and the second performance is underlined for both proprietary and open-source LMMs respectively.

| Categories LMM (*LLM*) | Technical↑ | Aesthetic↑ | Generative↑ | *Overall↑* |
|---|---|---|---|---|
| HUMAN (WORST) | 94.32% | 84.49% | 86.25% | 90.56% |
| HUMAN (BEST) | 94.69% | 86.01% | 93.00% | 92.22% |
| **Proprietary LMMs:** | | | | |
| GEMINI 1.5 PRO | **71.22%** | **77.61%** | 59.07% | **69.12%** |
| GPT-4V | 67.82% | 68.34% | 58.02% | 64.31% |
| GPT-4O (2024-05-13) | 70.59% | 61.61% | **67.92%** | 66.88% |
| QWEN-VL-MAX | 71.31% | 69.77% | 58.56% | 66.21% |
| **Open-source LMMs:** | | | | |
| Qwen2-VL-72B (*Qwen2-72B*) | **74.22%** | 60.31% | **70.23%** | **68.99%** |
| MiniCPM-V2.6 (*Qwen2-7B*) | 69.10% | 60.14% | 60.47% | 64.01% |
| InternVL2-40B (*Nous-Hermes-2-Yi-34B*) | 66.28% | 63.21% | 50.10% | 59.22% |
| Ovis1.5 (*Llama3-8B*) | 70.83% | 67.82% | 55.39% | 64.50% |
| LLaVA-OneVision (*Qwen2-7B*) | 68.84% | 67.79% | 54.27% | 63.78% |
| CogVLM2-19B (*Llama3-8B*) | 64.21% | 61.33% | 56.75% | 60.73% |
| IDEFICS-2 (*Mistral-7B-Instruct-v0.2*) | 62.00% | **68.76%** | 47.12% | 59.11% |
| DeepSeek-VL-7B | 55.91% | 53.79% | 47.59% | 52.36% |
| InternLM-XComposer2-VL (*InternLM2*) | 62.29% | 63.37% | 50.26% | 58.58% |
| LLaVA-NeXT (*Llama3-8B*) | 58.59% | 48.57% | 52.00% | 53.13% |
| LLaVA-NeXT (*Qwen-72B*) | 59.91% | 55.51% | 59.80% | 58.42% |
| LLaVA-NeXT (*Qwen-110B*) | 64.69% | 57.20% | 63.64% | 61.89% |
| mPLUG-Owl2 (*LLaMA-7B*) | 57.90% | 54.47% | 53.81% | 55.45% |
| LLaVA-v1.5 (*Vicuna-v1.5-7B*) | 45.90% | 41.33% | 54.59% | 47.12% |
| LLaVA-v1.5 (*Vicuna-v1.5-13B*) | 46.08% | 41.22% | 48.10% | 45.54% |
| CogVLM-17B (*Vicuna-v1.5-7B*) | 54.76% | 48.45% | 52.47% | 51.36% |
| Qwen-VL (*Qwen-7B*) | 49.46% | 34.34% | 50.49% | 44.99% |
| BakLLava (*Mistral-7B*) | 47.88% | 33.37% | 48.46% | 43.39% |
| Fuyu-8B (*Persimmon-8B*) | 44.61% | 30.23% | 45.65% | 40.20% |
| *random guess* | 31.87% | 32.92% | 33.14% | 32.63% |

sistent evaluations among humans, whereas aesthetic quality, being more subjective, results in a broader range of opinions and consequently lower performance scores.

**Human vs. Proprietary LMMs**   Proprietary (*closed-source*) LMMs are regarded as closely mirroring human perception and demonstrate superior performance, particularly in zero-shot settings for evaluating AIGI. Therefore, here we make a finer discussion about the human and proprietary LMMs. 1) Beginning with a detailed comparison of human and proprietary LMMs, we observe that proprietary LMMs achieve human-level performance in *Basic Recognition*, indicating their ability to correctly **assess AIGI alignment when prompts are simple**. 2) Despite this, LMMs encounter difficulties in the *Bag-of-Words* aspect, especially in identifying composition and counting objects, which highlights their **limitations in handling complex compositional relationships and specific object counts.** 3) In the *Outside Knowledge* domain, proprietary LMMs show only a slight performance gap compared to humans on *Specific Terms*, demonstrating comprehensive prior knowledge about specific terms, but they notably lag behind in identifying controversial content. While humans can easily recognize contradictory elements, proprietary LMMs often struggle due to their reliance on common sense, making accurate responses challenging. To conclude, according to the results shown in Table 1, proprietary LMMs are competent as evaluators for simple prompts in AIGI, yet they require further improvements for more complex prompts related AIGI content. 4) On the other hand, Table 2 reveals that LMMs have significant shortcomings in low-level quality perception compared to humans, with an uneven performance across different quality dimensions. Surprisingly, GPT-4o shows a distinct advantage over other proprietary LMMs in recognizing generative distortions, suggesting its superior capability in this area. However, the substantial overall difference in quality perception between proprietary LMMs and humans underlines that these models are currently **unsuitable for assessing the visual quality of AIGI**.

## 5   CONCLUSION

In conclusion, the ambition to employ LMMs for evaluating AIGIs exposes considerable deficiencies in their capabilities, as revealed by the diagnostic benchmark **A-Bench**. This benchmark scrutinizes

the core capabilities of LMMs themselves, focusing on their ability to accurately address fundamental questions related to high-level semantic understanding and low-level quality perception. Our findings from **A-Bench** serve as a stark reminder of the current limitations faced by LMMs in the realm of AIGI evaluation. The results underscore that while LMMs provide valuable insights, their **evaluation capacity remains notably inferior to human performance**, especially in tasks that demand deep semantic comprehension and detailed quality assessment. By identifying specific areas for enhancement and charting a course for future research, this study not only underscores the urgent need for further development but also aids in refining the application of LMMs in AIGI evaluation tasks. Future initiatives should focus on augmenting the capabilities of LMMs to reliably match or surpass human performance in these intricate evaluation scenarios.

## 6 ETHICS STATEMENT

This submission complies fully with the ethical guidelines set by ICLR 2025. We follow ICLR's principles for responsible AI development, ensuring that our research avoids any potential harm, bias, or discrimination. The data utilized in this work is sourced exclusively from publicly available open-source datasets and models. Furthermore, our methods prioritize fairness, accountability, and transparency in the evaluation of AI-generated images.

## 7 LIMITATIONS

**Timeliness Concern**   Creating a benchmark involves generating images, collecting data, training evaluators, and verifying data quality, making the process both time-consuming and costly. As a result, it is inevitable that AIGI benchmarks may not always keep pace with the latest technologies or models. However, the insights provided by the benchmark in evaluating AIGI remain valuable and offer useful guidance. We are committed to ongoing updates and expansions to ensure the benchmark remains current.

**Scale-up Concern**   Since the A-Bench dataset is fully manually annotated and requires validation by at least three other humans, the annotation process is both costly and time-consuming. As such, it is quite challenging to scale up.

## 8 ACKNOWLEDGEMENT

This work is supported in part by the National Natural Science Foundation of China (623B2073, 62101326, 62225112, & 62301316).

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

# A    APPENDIX

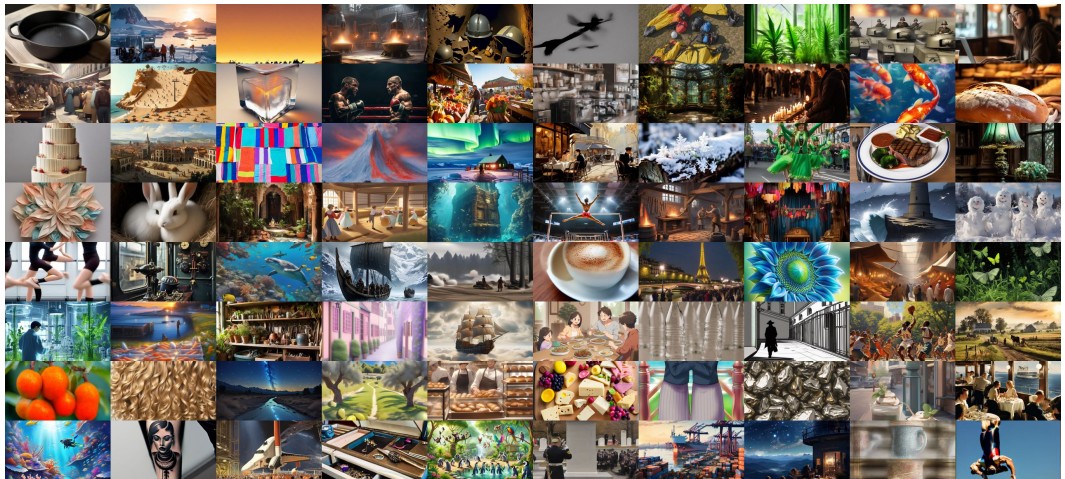

Figure 6: Overview of the AIGIs from **A-Bench**[P1].

## A.1    AIGIs COLLECTION

**AIGI collection for A-Bench**[P1]    To ensure that the AIGIs meet the specific subcategory requirements, we have gathered 2,000 manually-written prompts to serve as the textual foundation. Below, we provide examples of these prompts:

1. Basic Recognition ->Major Object Recognition: *An elaborate treehouse in a thick forest, with children playing inside, rope bridges connecting to other trees, and birds chirping around.*

2. Basic Recognition ->Minor Object Recognition: *A magical fairy ring in a moonlit forest, with tiny glowing fairies dancing and mystical plants all around.*

3. Bag-of-Words ->Attributes Awareness: *A delicate, frosty, crystal snowflake beside a warm, glowing, amber ember on a smooth, slate-gray stone.*

4. Bag-of-Words ->Nouns as Adjectives Awareness: *Shark-sleek submarine exploring ocean depths.*

5. Bag-of-Words ->Composition Identification: *A gamer's setup with consoles and controllers on a desk, multiple screens above, and game boxes and snacks partially obscured beneath the desk.*

6. Bag-of-Words ->Number of Objects Counting: *Six logs in a woodpile, stacked so tightly that they seem to form a solid block.*

7. Outside Knowledge ->Specific Terms Recognition: *A barometer showing a rapid decrease in pressure.*

8. Outside Knowledge ->Contradiction Overcome: *A ship floating above the clouds, sails made of sunlight.*

Afterward, we use the collected prompts to create AIGIs. **15** text-to-image generation models are selected, which include: Dreamlike (dreamlike art, 2023), Pixart $\alpha$ Chen et al. (2023a), Playground v2 PlaygroundAI (2023), SD1.4 (Rombach et al., 2022a), SD1.5 (Rombach et al., 2022a), SDXL (Rombach et al., 2022a), SSD1B (Gupta et al., 2024), LCM Pixart (Luo et al., 2023), LCM SD1.5 (Luo et al., 2023), LCM SDXL (Luo et al., 2023), SDXL Turbo Sauer et al. (2023) DALLE2 (Ramesh et al., 2022), DALLE3 (Ramesh et al., 2022), IF (DeepFloyd, 2023), Midjourney v5.2 Holz (2023). Finally, a total of 15×2,000 =30,000 AIGIs are collected. To guarantee diversity, we randomly select 2,000 AIGIs, choosing one AIGI per prompt. Subsequently, we conduct a manual review of these AIGIs to remove any that failed to generate correctly or are unsuitable for annotation. This process results in the final set of AIGIs for **A-Bench**[P1], which can be overviewed in Fig. 6.

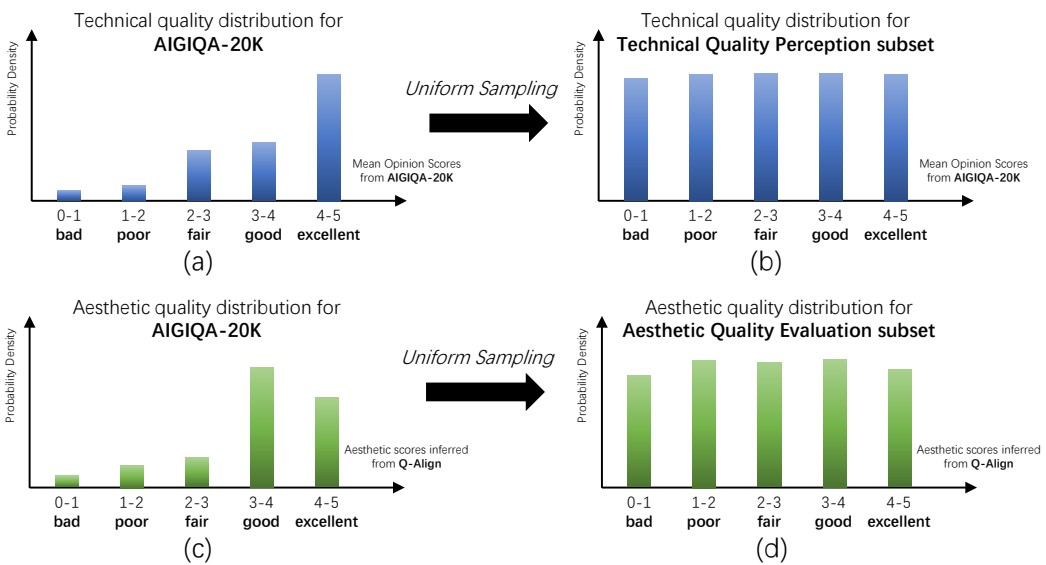

Figure 7: Illustration of the quality distribution transformation.

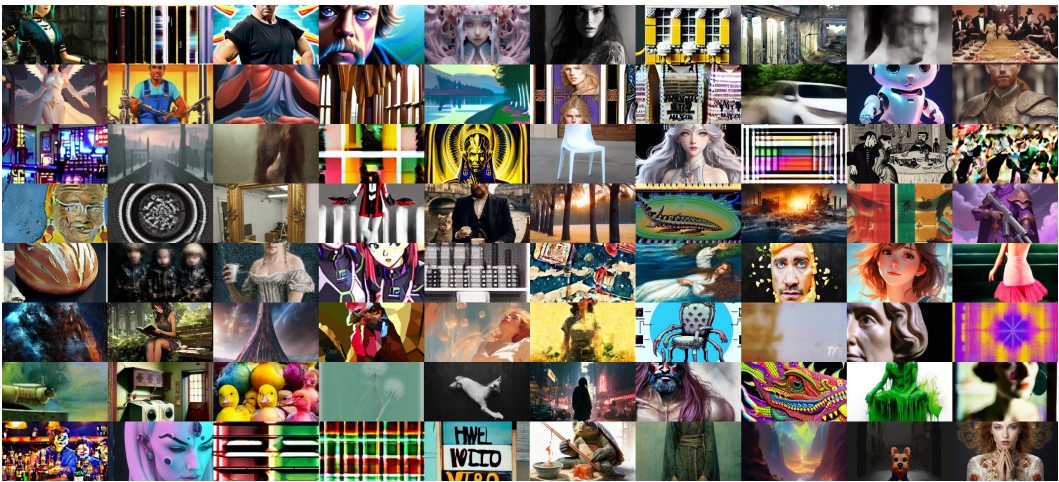

Figure 8: Overview of the AIGIs from **A-Bench**[P2].

**AIGI collection for A-Bench**[P2] **A-Bench**[P2] is designed for the quality evaluation of AIGIs. Consequently, it is essential to ensure that the collected AIGIs span a wide quality range to address various practical scenarios. For **Technical Quality**, we sample 500 AIGIs from the AIGIQA-20K dataset (Li et al., 2024) using a *uniform sampling* strategy. Specifically, each AIGI in the AIGIQA-20K dataset is assigned a mean opinion score (MOS) for technical quality. We apply uniform sampling to create more even distributions, as illustrated in Fig. 7. For **Aesthetic Quality**, in the absence of provided aesthetic scores, we utilize q-align (Wu et al., 2023c), an effective quality predictor, to infer the aesthetic values of AIGIs. Subsequently, we perform uniform sampling similarly to obtain 500 AIGIs for aesthetic evaluation. For **Generative Distortion**, we manually select 500 AIGIs exhibiting unexpected AIGI-specific distortions. It is important to note that there is no content overlap among the selected AIGIs, which can be overviewed in Fig. 8.

| Composition Identification -> Orientation | Composition Identification -> Orientation |
|---|---|
| What is located **to the left** of the desks in the classroom?
A. Educational posters on the walls (*correct*)
B. A teacher's desk
C. A blackboard
D. A bookcase | What is **partially covered** by cloaks hanging in the background of the magic workshop?
A. Shelves (*correct*)
B. Tricks and hats on a table
C. A magic wand
D. Cards spread out |
| **Composition Identification -> Size Comparison** | **Composition Identification -> Spatial Arrangement** |
| In the paleontologist's dig site, which seems to be **the largest**?
A. Human on the right
B. Human in the middle
C. The field journal (*correct*)
D. Pens | Are strategy boards placed **in all four corners**?
A. No (*correct*)
B. Yes |
| **Specific Terms -> Geography** | **Specific Terms -> Brand** |
| What **geography feature** is depicted here?
A. Sandy beach
B. Mangrove forest
C. Coral reef
D. Rugged coastline (*correct*) | **Which brand** is famous for this item?
A. Shell
B. Sony
C. Nike (*correct*)
D. Amazon |
| **Specific Terms -> Food** | **Specific Terms -> Style** |
| What is the **main cooking technique** used for the meat in this dish?
A. Poaching
B. Frying
C. Grilling (*correct*)
D. Roasting | What **painting style** is represented in the image?
A. Baroque
B. Rococo (*correct*)
C. Neoclassicism
D. Art Nouveau |

Figure 9: Some finer cases for the 'Bag-of-Words ->Composition Identification' and 'Outside Knowledge ->Specific Terms' subcategories.

## A.2 FINER EXPLANATION FOR SOME SUBCATEGORIES

For certain subcategories that require additional clarification for better understanding, we provide detailed explanations here (the corresponding cases are shown in Fig. 9):

1. Bag-of-Words ->Nouns as Adjectives Awareness. The 'Noun as Adjectives' illustrates the use of nouns as adjectives to modify objects in AIGIs. Essentially, we aim for the descriptive effect, not for the nouns themselves to be visually represented in the AIGIs. For instance, as shown in Fig.4 row 2 column 2, when we describe a submarine as 'shark-sleek,' we do not intend to generate an image of an actual shark. This subcategory is designed to test whether LMMs can correctly identify such misunderstandings.

2. Bag-of-Words ->Composition Identification. We categorize composition into four distinct types: 1) Orientation, which assesses the ability to correctly determine the relative spatial positions of objects; 2) Occlusion, which involves evaluating the accuracy in discerning the overlapping relationships between objects; 3) Size Comparison, which tests the ability to accurately judge the size relationships among objects; and 4) Spatial Arrangement, which examines the ability to accurately assess the arrangement of objects within the AIGI.

3. Outside Knowledge ->Specific Terms. This subcategory covers many aspects, including geography, sports, science, materials, food, everyday life, creatures, brands, and styles. This primarily investigates whether it is possible for LMMs to infer and deduce specific knowledge within these fields based on the content of AIGIs such as identifying the exact location feature based on geographical attributes, deducing the brand from the characteristics of a product, recognizing the cooking technique of the food, etc.

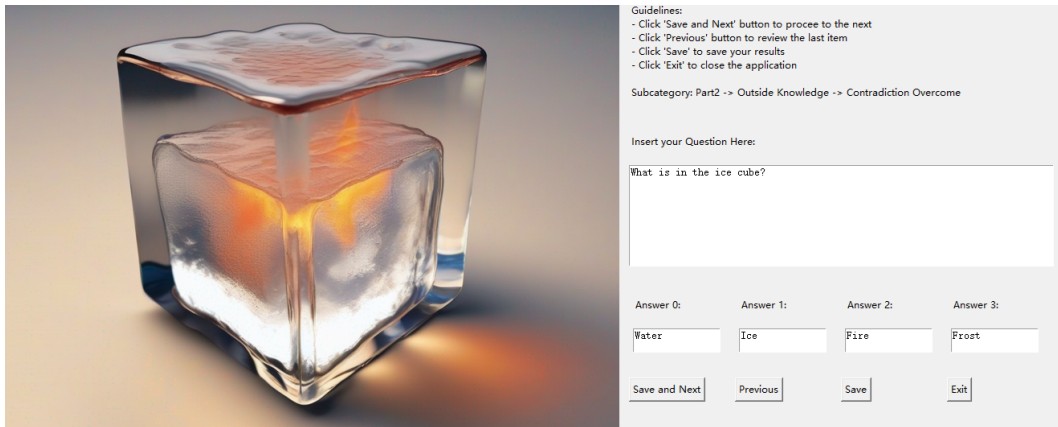

Figure 10: Illustration depicting the annotation interface, where experts are presented with the sub-category and are able to record their questions and answers.

### A.3 HUMAN EXPERT ANNOTATION

A total of fifteen experts, each possessing professional skills and extensive experience in photography and AIGIs, participate in the subjective labeling experiment of **A-Bench**. All experts are informed that their annotation data will be publicly released, and they all agree to this arrangement. The hourly wage for each expert is approximately 12 US dollars, resulting in a total expense of about 2,400 US dollars for the whole subjective experiment.

The experiment takes place in a laboratory environment with standard indoor lighting. A Dell 4K monitor, supporting a resolution of $3840 \times 2160$, is used for displaying the interfaces. Screenshots of the interfaces can be referred to in Fig. 10. Each expert annotates up to 30 AIGIs per day to avoid fatigue, with every annotation carefully reviewed by at least three other experts before acceptance. This approach ensures the highest possible accuracy and rigor of the **A-Bench** labels, thereby enhancing the precision and meaningfulness of the performance testing capability of **A-Bench**.

### A.4 GPT EVALUATION FOR CHOICE JUDGMENT

For some LMMs, the response to the question inquiry may vary. For example, given the correct answer *C. Blurry*' to the question *What is the most severe technical distortion of this image?*', LMMs may respond in different formats: *The image is blurry*', *There is blur in this image*', or '*low clarity*'. To address the impact of such variations on our evaluation, we've implemented a 5-round **voting** strategy (Wu et al., 2023a). Under this strategy, we pose the same prompt, as defined in the templates, five times and determine the final outcome based on the majority of GPT's responses.

**GPT Evaluation Prompt Template**

*#System: You are a helpful assistant that grades answers related to image perception. There are a lot of special terms or keywords related to image processing and photography.*

*#User: Assuming you are a grader, you will now be provided with a question [question] and a set of options [options] with option [options[0]] being the correct answer. Additionally, there will be an answer [answer] provided by a respondent. Please determine whether the respondent's answer is correct considering the context of the question. Even if the word choice is not completely the same, you can decide based on the given options and see whether the one in the answer is close enough to the given correct answer, The result is 1 if the answer is correct and else the result is 0. Please only provide the result in the following format: Result:*

**Example for GPT Evaluation**

**Question:** Which is the most blurry part of this image?

**Choices:** ['The house on the left', 'The person in the middle', 'The background', 'The tree on the left']

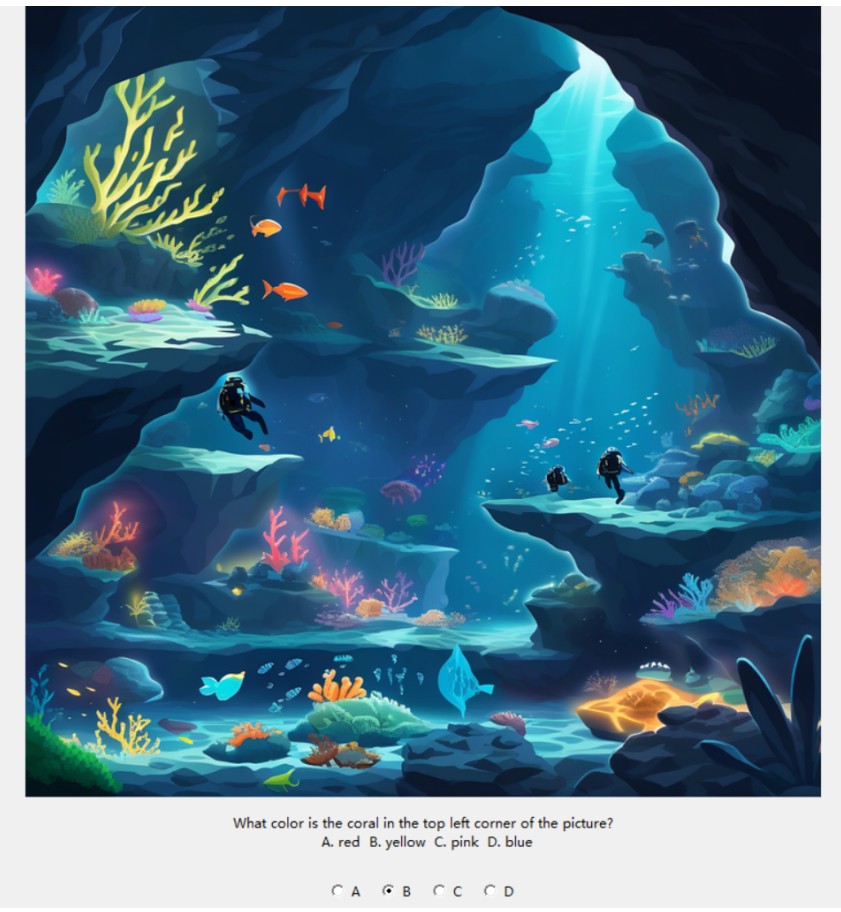

Figure 11: Illustration of the interface for the user-study.

**LMM Answer:**

The most blurry part in this image is the house to the left of the person.

***5-Round GPT Answers:***
*[“Score: 1”,“Score: 1”,“Score: 1”,“Score: 1”,“Score: 1”]*
$\rightarrow$ Final Correctness after Voting: ✓

### A.5    USER-STUDY ON A-BENCH

To provide human performance on the **A-Bench**, we employ five ordinary people in a controlled laboratory setting. Initially, participants familiarize themselves with the tasks through exposure to similar cases. Subsequently, they select the appropriate responses for the questions posed in the A-Bench. The user-study interface is shown in Fig. 11.

### A.6    LMM EXPERIMENT DETAILS

The LMMs undergo testing in a zero-shot setting. Proprietary LMMs are evaluated via official APIs, whereas the *open-source* LMMs (with the exceptions of LLaVA-NeXT *Qwen-72B* and LLaVA-NeXT *Qwen-110B*) run on an NVIDIA RTX 6000 Ada with 48 GB of memory. The LLaVA-NeXT *Qwen-72B* and LLaVA-NeXT *Qwen-110B* are operated on 4 NVIDIA H100 with 320 GB of memory. All LMMs operate with default parameters, ensuring that the **A-Bench** results are readily reproducible.

Table 3: Benchmark results on the question types. The best performance is marked in **bold** and the second performance is underlined for both proprietary and open-source LMMs respectively.

| Categories | A-Bench$^{P1}$ | | A-Bench$^{P2}$ | | Overall | |
|---|---|---|---|---|---|---|
| **LMM** (*LLM*) | *Yes-or-no*↑ | *What*↑ | *Yes-or-no*↑ | *What*↑ | *Yes-or-no*↑ | *What*↑ |
| HUMAN (WORST) | 91.21% | 92.77% | 89.45% | 91.02% | 91.23% | 91.88% |
| HUMAN (BEST) | 93.55% | 94.25% | 91.80% | 92.64% | 92.77% | 93.39% |
| **Proprietary LMMs:** | | | | | | |
| GEMINI 1.5 PRO | 81.96% | **86.91%** | **74.08%** | **65.57%** | **76.50%** | **76.82%** |
| GPT-4V | 82.37% | 85.86% | 71.11% | 60.09% | 75.51% | 73.23% |
| GPT-4O | 84.39% | 85.76% | 69.76% | 65.15% | 76.28% | 75.81% |
| QWEN-VL-MAX | **86.70%** | 84.02% | 68.13% | 64.60% | 75.79% | 74.91% |
| **Open-source LMMs:** | | | | | | |
| CogVLM2-19B (*Llama3-8B*) | 81.77% | 83.26% | 63.70% | 58.65% | 70.55% | 71.61% |
| IDEFICS-2 (*Mistral-7B-Instruct-v0.2*) | 78.32% | 83.84% | 63.87% | 55.63% | 68.91% | 69.96% |
| DeepSeek-VL-7B | 80.72% | 82.00% | 60.00% | 47.15% | 66.88% | 66.48% |
| InternLM-XComposer2-VL (*InternLM2*) | 82.08% | 81.53% | **66.49%** | 53.06% | 70.90% | 69.83% |
| LLaVA-NeXT (*Llama3-8B*) | 81.17% | 84.11% | 52.10% | 53.77% | 63.89% | 68.82% |
| LLaVA-NeXT (*Qwen-72B*) | **83.22%** | **84.31%** | 57.91% | 60.01% | 70.22% | 71.55% |
| LLaVA-NeXT (*Qwen-110B*) | 82.99% | 83.91% | 59.78% | **62.87%** | **71.76%** | **73.05%** |
| mPLUG-Owl2 (*LLaMA-7B*) | 74.92% | 78.00% | 56.97% | 54.36% | 64.38% | 67.81% |
| LLaVA-v1.5 (*Vicuna-v1.5-7B*) | 78.27% | 82.74% | 46.39% | 47.97% | 58.85% | 66.21% |
| LLaVA-v1.5 (*Vicuna-v1.5-13B*) | 79.51% | 81.47% | 47.23% | 43.90% | 61.41% | 63.61% |
| CogVLM-17B (*Vicuna-v1.5-7B*) | 76.77% | 80.11% | 55.13% | 49.71% | 64.33% | 65.65% |
| Qwen-VL (*Qwen-7B*) | 72.77% | 80.95% | 46.22% | 44.02% | 56.60% | 63.39% |
| BakLLava (*Mistral-7B*) | 71.01% | 78.77% | 42.11% | 44.11% | 55.61% | 60.03% |
| Fuyu-8B (*Persimmon-8B*) | 61.56% | 64.22% | 38.76% | 41.66% | 50.06% | 52.31% |

## A.7 QUESTION TYPE PERFORMANCE

We assess the performance disparity between *Yes-or-no* and *What* questions among LMMs. The *Yes-or-no* questions gauge the fundamental judgment capabilities of LMMs, whereas *What* questions demand a more comprehensive understanding. According to the results in Table 3, it is observed that most LMMs perform better on *What* questions within **A-Bench**$^{P1}$, suggesting a proficiency in processing semantic content. Conversely, in **A-Bench**$^{P2}$, where LMMs generally show lesser performance, they exhibit limited in-depth perception, maintaining only basic evaluative capabilities without comprehensive understanding, leading to poorer performance on *What* questions. Interestingly, human performance consistently excels in *What* questions across both **A-Bench**$^{P1}$ and **A-Bench**$^{P2}$, likely due to a broader range of options facilitating easier inference. However, human performance tends to be more balanced compared to LMMs, which may exhibit significant variance, such as IDEFICS-2, where there is over a 5% accuracy difference between question types, indicating less robustness.

## A.8 RESPONSE VARIANCE FOR LMMS

Considering that the accuracy and stability of the benchmark directly affect the quality of the evaluation, therefore we conduct the response variance experiment here. First, we use a consistent prompt instruction format to minimize any misunderstanding by LMMs and standardize the output. Additionally, we set the model's temperature parameter to 0, meaning the LMM's output will no longer be affected by randomness. As a result, the model will give the same response to the same question each time, eliminating variance.

It's also worth noting that increasing the model's temperature to encourage more diverse and exploratory answers is indeed an interesting consideration. To further address the concern about the statistical significance of the experiment, we repeat the A-Bench experiment for 5 rounds with different temperature settings across several popular 7B-8B LMMs. The performance is listed in the table below, with the results presented as the mean accuracy ± standard error.

Based on the results, we can observe that when the temperature is set to zero, the accuracy results for all LMMs remain consistent across all 5 rounds. As the temperature increases, the average

| Temperature | DeepSeek-VL-7B | LLaVA-NeXT-8B | LLaVA-v1.5-7B | Qwen-VL-7B |
|---|---|---|---|---|
| 0.0 | 66.58±0.00 | 67.75±0.00 | 62.97±0.00 | 60.41±0.00 |
| 0.5 | 65.11±1.72 | 66.43±2.09 | 60.61±2.23 | 58.17±1.89 |
| 1.0 | 62.04±4.51 | 63.77±3.86 | 59.22±4.01 | 55.22±6.04 |

Table 4: Performance comparison at different temperatures for various LMM models.

performance declines and the results become more unstable, with higher standard errors. Therefore, to ensure reproducibility and performance stability, we prefer the zero-temperature setting, as it more accurately and reliably reflects the performance of LMMs, making it more suitable for practical applications.

## A.9    DATA STATEMENT

The **A-Bench** dataset is released under the **CC BY 4.0** license. This includes all associated AIGIs, questions, and answer candidates. However, to prevent incorporation into the training sets of any LMMs, the correct answers remain confidential. We believe this precaution will ensure that **A-Bench** retains its long-term value as a benchmark for assessing AIGI evaluation capabilities.

## A.10    FURTHER LIMITATIONS AND SOCIAL IMPACT

**Limitations**    While **A-Bench** uses a diverse set of generative models and LMMs for evaluation, the choice and number of models might still limit the generalizability of the results. The performance of untested models or newer generative approaches might differ significantly. The rapid advancement in AI and generative models may quickly outpace the current setup of A-Bench, necessitating frequent updates or redesigns of the benchmark to stay relevant.

**Social Impact**    By improving the evaluation metrics for AIGIs, **A-Bench** could lead to more reliable and trustworthy AI-generated content, which is crucial as these technologies increasingly intersect with areas like media, entertainment, and education. Moreover, improved benchmarks and evaluation methods can drive industry standards, potentially lowering the barrier to entry for smaller developers and promoting innovation through clearer performance targets.

