# OpenReview forum: "A-Bench: Are LMMs Masters at Evaluating AI-generated Images?"
_ICLR.cc/2025/Conference — ICLR 2025 Poster_

### Official Review · Reviewer_AWPs · 2024-10-30

**Soundness:** 3
**Presentation:** 4
**Contribution:** 3
**Rating:** 6
**Confidence:** 4

**Summary:**

This paper study the multimodal LLM's ability in the context of image evaluation. Instead of studying the effectiveness of certain LLM-based metrics, This work aims to identify whether multimodal LLM (LMM) are truly capable of evaluating AI-generated images through a question answering benchmark. Proposed a benchmark that contains 2,864 set of questions which can be categorized into 6 categories, involving semantic understanding and quality perception. After benchmarking a total of 18 LMMs and comprehensive analysis, the authors came up with a conclusion that LMMs are still not masters at evaluating AI-generated images.

**Strengths:**

S1) The paper is well-written and well-organized. All the relevant details are included in the main paper and appendix. The evaluating method also seems rigorous.

S2) A good pitch in studying LMMs directly through question answering instead of studying the effectiveness of certain LMM-based metrics. This helped to shed a light on the true capabilities of current LMM-based image evaluation metrics.

**Weaknesses:**

W1) The paper would have provided more insights if the authors also studied the reasoning to verify if the LMMs truly understand how to evaluate for each categories (i.e. did the reasoning fully explain the choice made by the LMM?). This might help to explain the gap between the performance of LMMs and humans. I suggest conducting a study on small subset for each categories and see how the reasoning was aligned to the choice made.

W2)  It would also be desirable to see what kind of images LMM evaluate poorly across each categories in AIGI. A more detailed of diversity analysis on the AIGI dataset is required. E.g. for Basic Recognitions, how much portion of the questions are regarding recognitions of animal, human, or artifacts? Are these LMMs doing poorly on particularly certain type of objects?

**Questions:**

Q1) What is SRCC/PLCC in the introduction paragraph? It seems this abbreviation is never explained in the paper.

Q2) In section 4.1, "It’s worth noting that the instruction prompt might slightly differ for different LMMs according to the official setting." Why the instruction prompt is slight different for different LMMs? How will it impact the performance of LMMs?

Q3) For the human annotators, how are they recruited? What kind of training were they given? How many instances are labelled by each human annotator? I am also interested in the total time required to build this benchmark.

---

> ### Author Response · Authors · 2024-11-17
> **Official Response to Reviewer AWPs**
>
> First, we would like to thank the reviewer for the constructive and valuable feedback. We have addressed your concerns point-by-point below.
>
> **1. How LMMs Reason Before Giving the Choice**
>
> Thanks for pointing out the importance of verifying whether LMMs truly understand their choice. Following your suggestions, we randomly sample a small subset which consists of 200 question-answer pairs for validation. Specifically, we modify the question query into:
>
> ```
> #User: [Question] [Image Token]
> [Choice A] [Choice B] [Choice C] [Choice D]
> Answer with the option’s letter from the given choices and give the reasons for why you choose this option.
> ```
>
> The prompt includes the instruction **give the reasons for why you choose this option** to encourage the LMMs to explain their choices, allowing us to assess whether they truly understand the question or are merely guessing. Considering the complexity of LMM responses, introducing additional LMMs as judges could introduce bias; therefore, we conducted a **user study to evaluate the alignment of the LMMs' reasoning processes with their choices.**
> This evaluation is only applied when the LMMs provide the correct option. Specifically, we present observers with the image, question-answer pair, and the corresponding LMM response, asking them to judge whether the LMM’s choice and reasoning are aligned. If aligned, the response is scored 1, if not, it is scored 0. Each evaluation involves two observers. If their judgments match, the result is recorded, if they disagree, a third observer arbitrates, and the majority decision is recorded. We select some LMMs (GPT-4o, Gemini 1.5 Pro, CogVLM2-19B, LLaVA-NeXT-8B) with good performance on A-Bench for this experiment. The results are illustrated below:
>
> | LMM             | Basic Recognition | Bag-of-Words | Outside Knowledge | Technical | Aesthetic | Generative | Overall |
> |-----------------|-------------------|--------------|-------------------|-----------|-----------|------------|---------|
> | GPT-4o          | 0.95              | 0.93         | 0.93              | 0.82      | 0.85      | 0.80       | 0.90    |
> | Gemini 1.5 Pro  | 0.94              | 0.92         | 0.91              | 0.87      | 0.81      | 0.79       | 0.89    |
> | CogVLM2-19B     | 0.93              | 0.93         | 0.91              | 0.82      | 0.79      | 0.74       | 0.86    |
> | LLaVA-NeXT-8B   | 0.92              | 0.94         | 0.92              | 0.81      | 0.81      | 0.76       | 0.86    |
>
> From the results, we observe that in terms of semantic understanding, LMMs generally grasp the content of the questions well, with their reasoning process aligning closely with the chosen options. However, in areas related to visual quality understanding—where their performance is already weaker—some correct answers do not align with the reasoning, indicating a few cases of correct guesses. Nevertheless, the overall alignment accuracy remains acceptable, and the LMMs show relatively stable performance across questions within the same category. This suggests that A-Bench testing is still reasonable, accurate, and meaningful.
>
>
> **2. LMMs' Sensitivity to Content**
>
> Thanks for your comment. We further conduct a content sensitivity experiment based on your suggestions. We split the content into **Human**, **Animal**, and **Objects**, then we calculate the performance of each content type for illustration.
>
> | LMM             | Human | Animal | Objects |
> |-----------------|-------------------|--------------|--------------|
> | GPT-4o          | 0.71              | 0.66         | 0.80             |
> | Gemini 1.5 Pro  | 0.70              | 0.63         | 0.79            |
> | CogVLM2-19B     | 0.68              | 0.64         | 0.74          |
> | LLaVA-NeXT-8B   | 0.60              | 0.55         | 0.69         |
>
> Interestingly, our findings indicate that LMMs perform best on images with **Objects** content, while they struggle more with images featuring **Animal** content. This may be because objects tend to be simpler and more straightforward, making them easier for the models to understand. In contrast, animals often exhibit unusual or complex generated structures and patterns, which can negatively impact the accuracy of LMMs' understanding.

---

> > ### Author Response · Authors · 2024-11-17
> > **Official Response to Reviewer AWPs**
> >
> > **3. Missing Definition for SRCC/PLCC**
> >
> > We apologize for the missing definitions of **SRCC** and **PLCC** and appreciate you bringing this to our attention. Spearman's Rank Correlation Coefficient (SRCC) and Pearson's Linear Correlation Coefficient (PLCC) are widely used metrics for evaluating the correlation between predicted scores and ground truth scores in tasks like quality assessment.
> >
> > **SRCC** measures the rank-based correlation between two variables. Instead of comparing the raw values, it assesses the relationship based on the relative ordering (ranks) of the scores.
> >
> > **PLCC** measures the linear relationship between two continuous variables. Unlike SRCC, it uses raw values rather than ranks, calculating how well one set of scores can predict another through a linear relationship.
> >
> > **4.  Different Instruction Prompts for Different LMMs**
> >
> > Thank you for your question， we’re happy to clarify. The default prompt used in A-Bench may cause certain LMMs to struggle with selecting the correct option. For instance, when testing BakLLava-7B, it occasionally outputs irrelevant text instead of selecting from the options. However, when we modify the prompt from **Answer with the option’s letter directly from the given choices** to **Please tell me the choice for the correct answer**, these errors occur much less frequently. Through multiple trials, we found that while our default prompt generally provides stable instructions for LMM responses, some LMMs occasionally need slight adjustments. These modifications are minor and infrequent, so their overall impact on results is minimal.
> >
> >
> > **5. Detailed Information About Human Annotation**
> >
> > Thank you for your question.
> > First, we issue an invitation to recruit participants familiar with visual quality and AIGI for in-person training. During the training, participants are introduced to the annotation tasks they will perform. They then practice with additional AIGI data prepared for hands-on annotation, following specific requirements. Afterward, we organize expert-led discussions and assessments of the annotations. Those who pass the evaluation are recruited as annotators. Ultimately, fifteen participants successfully complete the training and are selected.
> >
> > To prevent fatigue, each person is limited to annotating a maximum of thirty entries per day. Each annotated entry has to be reviewed and approved by three other participants for it to be considered valid (with each review also counted as an annotation). On average, each person annotates approximately 750 entries, and the entire annotation process takes about two months to complete.
> >
> > Thanks again for your comments on refining our work.

---

> > > ### Comment · Reviewer_AWPs · 2024-11-21
> > > **Response to rebuttal.**
> > >
> > > Hi, thanks for the detailed response. I will keep the score as it is.

---

### Official Review · Reviewer_vuEQ · 2024-11-03

**Soundness:** 4
**Presentation:** 4
**Contribution:** 3
**Rating:** 8
**Confidence:** 5

**Summary:**

- Human evaluations are the gold standard for evaluating generative models especially text-to-image (T2I) models. However, they are expensive. An alternative is using automatic metrics and Large Multi-modal Models (LMMs) are a popular choice.
- LMMs are trained on real images and AI-generated Images (AIGIs) are out of domain for LMMs questioning their reliability as evaluation models.
- This work proposes A-Bench a diagnostic benchmark for assessing the reliability of LMMs for evaluating AIGIs.
- A-Bench consists of two subsets 1) A-Bench P1 to evaluate the text faithfulness or prompt adherence of T2I models and 2) A-Bench P2 to evaluate the quality of the generations.
- Authors samples 2864 AIGIs from 16 open and closed source T2I models. For each generation, they sourced human experts to annotate question-answer pairs and computed the accuracies of popular proprietary and open-source LMMs.
- Authors report that 1) Proprietary LMMs are better than open-source counterparts for text faithfulness, 2) proprietary LMMs perform as well as humans on simple enough prompts and 3) LMMs are not good models to evaluate the generation quality of AIGIs.

**Strengths:**

- Authors address a very important problem: Are current LLM/LMMs good enough to be used as judges for generative models? This line of research can provide valuable insights to train better LMMs for understanding AIGIs.
- A-Bench along with standard LMM evaluation benchmarks provide a complete picture of an LMMs capability to understand both real and AI generated images.
- The paper is well written and very easy to follow containing all the details necessary for reproduction.
- The experimental section is exhaustive with comparisons provided for both proprietary and open-source LMMs.

**Weaknesses:**

- I didn't find any major weakness with this work.

**Questions:**

- I would recommend authors check out SelfEval [1] as another source of evidence that external models cannot be reliable for evaluating T2I models. Please discuss it if relevant.
- In my experience there is a huge variance to the responses provided by  LLMs/LMMs. Did the authors compute variance of the scores or perform any statistical significance studies?
- L272 controversial -> counter factual
- In the introduction (L112-L117), in my opinion, authors should provide some numbers to make the point that LMMs are still not masters at evaluating AIGIs. Right now authors state that "there remains a considerable gap and significant room for improvement". Instead providing some numbers can make it more straightforward.

[1] Sai Saketh Rambhatla, Ishan Misra, SelfEval: Leveraging the discriminative nature of generative models for evaluation

---

> ### Author Response · Authors · 2024-11-17
> **Official Response to Reviewer vuEQ**
>
> First and foremost, we would like to thank the reviewer for the time and valuable feedback. We are sincerely grateful for the recognition and appreciation expressed. Our point-by-point responses are as follows:
>
> **1. Discussion About SelfEval**
>
> Thanks for your suggestions. We carefully review SelfEval [1] and find it highlights key weaknesses of using external models, such as: ``Evaluation metrics can vary widely depending on the chosen model, impacting reliability. If the same model is used in both training and evaluation, results may be biased, not reflecting true performance. External models often struggle with certain tasks, such as counting or recognizing specific attributes, making their evaluation scores unreliable'' This provides strong evidence that external models may not be reliable for evaluating T2I models.**We will include a discussion of SelfEval in the introduction.** Thank you for the reminder.
>
> **2. Responses Variance of LLMs/LMMs**
>
> Thank you for your question. Your concern is crucial, as the accuracy and stability of the benchmark directly affect the quality of evaluation. Here, we will address this:
> First, we use a consistent prompt instruction format to minimize any misunderstanding by LMMs and standardize the output. Additionally, we set the model's temperature parameter to 0, meaning the LMM's output will no longer be affected by randomness. As a result, the model will give the same response to the same question each time, eliminating variance.
>
> It’s also worth noting that increasing the model's temperature to encourage more diverse and exploratory answers is indeed an interesting consideration. **To further address your concern about the statistical significance of the experiment, which we believe is crucial and important, we repeat the A-Bench experiment for 5 rounds with different temperature settings across several popular 7B-8B LMMs.** The performance is listed in the table below, with the results presented as the mean accuracy ± standard error.
>
> | Temperature | DeepSeek-VL-7B | LLaVA-NeXT-8B | LLaVA-v1.5-7B | Qwen-VL-7B  |
> |-------------|----------------|---------------|---------------|-------------|
> | 0.0         | 66.58±0.00     | 67.75±0.00    | 62.97±0.00    | 60.41±0.00  |
> | 0.5         | 65.11±1.72     | 66.43±2.09    | 60.61±2.23    | 58.17±1.89  |
> | 1.0         | 62.04±4.51     | 63.77±3.86    | 59.22±4.01    | 55.22±6.04  |
>
> Based on the results, we can observe that when the temperature is set to zero, the accuracy results for all LMMs remain consistent across all 5 rounds. As the temperature increases, the average performance declines and the results become more unstable, with higher standard errors. Therefore, to ensure reproducibility and performance stability, we prefer the **zero-temperature setting**, as it more accurately and reliably reflects the performance of LMMs, making it more suitable for practical applications.
>
> **3. Inappropriate Words and Writing Improvement**
>
> Thanks for your constructive suggestions. We have changed the word `controversial` on L272 into  `counter factual`.
> We have improved our statement in the introduction that `A substantial **performance gap of 16%** remains between the best-performing LMMs and human evaluators on AIGI assessments, indicating significant room for improvement.'
>
> Thanks again for your valuable suggestions on improving our work.
>
> [1] Sai Saketh Rambhatla, Ishan Misra, SelfEval: Leveraging the discriminative nature of generative models for evaluation

---

> > ### Comment · Reviewer_vuEQ · 2024-11-23
> > **Response to author comments**
> >
> > I thank the authors for their detailed response.
> > The variance analysis should be included in the paper.
> > Overall I think this is a good paper and research in this direction is warranted given the exponential progress in generative models and their capabilities. I vote to keep my ratings.

---

### Official Review · Reviewer_WKMw · 2024-11-07

**Soundness:** 2
**Presentation:** 2
**Contribution:** 2
**Rating:** 5
**Confidence:** 3

**Summary:**

The paper introduces a benchmark designed to assess the efficacy of large language models (LLMs) in evaluating AI-generated images (AIGI). As the field increasingly depends on LLMs for this evaluation—sidestepping the high costs and time commitments of traditional user studies—quantifying the quality and reliability of LLM-based assessments is essential. While it's generally accepted that LLM evaluations fall short of human assessments, this paper provides a systematic analysis of the performance gap across various LLMs, comparing open-source and closed-source models to human evaluations.

The benchmark defines several key metrics within two primary dimensions: Semantic Reasoning and Quality Perception. Using this framework, the study measures the performance of multiple LLMs, revealing a substantial disparity between human judgment and LLM performance in AIGI evaluation.

**Strengths:**

1. The benchmark is undoubtedly useful. Given the growing reliance on LLMs to evaluate various AI-generated content like images, having a comprehensive, quantitative benchmark that assesses the effectiveness of LLMs in evaluation is highly valuable.
2. The paper tries to objectively define the underlying metrics of evaluation.
3. The benchmark development involved a rigorous process, starting with user studies to establish a baseline, followed by testing various LLMs, which adds credibility and depth to the analysis.
4. While the findings align with expectations, quantifying the gap between human and LLM performance is a valuable contribution. It enables the research community to approach improvements in this field with a more data-driven perspective, facilitating measured, progressive advancements.

**Weaknesses:**

1. While the metrics cover several important facets of semantic reasoning, they lack a rigorous scientific foundation, raising questions about whether they capture the full scope of semantic understanding as implicitly perceived by humans. Specific dimensions of semantic reasoning, such as cultural nuances, or emotional depth, may be missing from the current metrics, which could impact the holistic evaluation of AI-generated images. As such, while the comparisons of different LLMs using these metrics provide intriguing insights, it remains questionable whether these metrics are robust enough to serve as a truly holistic benchmark for evaluating semantic reasoning in AI-generated images.

2. The number of images used (~2,000) feels arbitrary and may be insufficient to capture the nuanced aspects of reasoning and quality perception required for a comprehensive evaluation. Expanding the dataset to around 5,000–10,000 images, with careful attention to diversity across image types and contexts, could improve the robustness of the analysis. Additionally, it would be helpful for the authors to provide a rationale for this dataset size or acknowledge any limitations they faced in scaling up.

**Questions:**

1. It would strengthen the paper if the authors could provide scientific backing for the proposed metrics, citing sources that systematically define each measure. Specific areas where additional references might be valuable include the validity of semantic reasoning components and quality perception dimensions, to help ensure that the chosen metrics align with established frameworks in the field.

2. Providing further details on the image selection process would clarify the robustness of the benchmark. Specifically, information on the criteria for image selection, the diversity of image types, and the distribution of different content categories would offer valuable context. If possible, outlining how these factors impact the benchmark's representativeness, and validity could further enhance transparency.

---

> ### Author Response · Authors · 2024-11-17
> **Official Response to Reviewer WKMw**
>
> First of all, we would like to thank the reviewers for the time and constructive feedback. We will address your concerns point by point.
>
> **1. Scientific Foundations for the Semantic Reasoning**
>
> We appreciate your inquiries about the scientific foundations of our selected aspects. We agree that ensuring comprehensive coverage is essential. When evaluating the semantic capabilities of LMMs, it is generally recommended to test from simpler to more complex tasks [1]. Thus, our aspect selection **follows a progression from basic to more complex dimensions**. Since our primary focus is on exploring LMM-based AIGI evaluations, we draw on previous work that uses LMMs for evaluation metric design [2, 3, 4], where LMMs are typically employed for image-related question-answer tasks.
> To evaluate the alignment between an AIGI and its prompt, we first assess overall subject alignment, which led us to select the first dimension, `Basic Recognition`. We then evaluate the attributes, actions, and interactions of objects in the AIGI, leading to the second dimension, `Bag-of-Words Pitfalls Discrimination`. Finally, given the creative nature of AIGI generation, which often requires external knowledge, we chose the third dimension, `Outside Knowledge Realization`.
>
> We further refined our dimensions by considering specific aspects of AIGI generation that are particularly relevant:
>
> 1. `Major Object Recognition` and `Minor Object Recognition` focus on identifying generated objects, which is a fundamental capability of AIGIs [5, 6].
> 2. `Attributes Awareness` evaluates the model's sensitivity to object attributes, which is crucial for basic evaluations [1, 7].
> 3.  `Nouns as Adjectives Awareness` addresses potential issues where T2I models may misinterpret nouns as adjectives, generating objects instead of intended attributes [8, 9].
> 4. `Composition Identification` pertains to understanding compositional relationships [10, 11].
> 5. `Number of Objects Counting` assesses the model's ability to accurately count objects, which is critical for checking if the AIGI matches numerical specifications in the prompt [12, 13].
> 6. `Specific Terms Recognition` involves identifying domain-specific scenes and objects, such as geography, sports, or food, important for external knowledge [14].
> 7. `Contradiction Overcome` tests the model's ability to correctly interpret AIGIs even when their content contradicts established world knowledge [15].
>
> Thus, our dimension selection follows **a suggested benchmark approach from simple to complex, with sub-dimensions designed to address critical points in AIGI generation and LMM evaluation.** While we cannot guarantee absolute comprehensiveness in the semantic domain, we have aimed to cover the most important aspects for evaluating AIGI capabilities with LMMs. In line with your suggestions, **we have provided corresponding citations and restructured our framework to align with the current evaluation setup.** Thank you for your thoughtful feedback.
>
> **2. About the Dataset Size**
>
> Thank you for your comment. To begin with, we would like to briefly outline the dataset sizes of some of the poular LMM evaluation benchmarks. For example, the **MMBench** benchmark, which is widely recognized for its comprehensive evaluation of semantic understanding abilities, contains **2,948 multiple-choice questions (MCQs)** [1]. In the domain of perceptual quality, the Q-Bench benchmark, which is quite popular, includes around **2,990 MCQs** [16]. Based on these examples, we aimed to keep our dataset size around 3,000 MCQs. Initially, we designed a total of 3,000 MCQs, but during the data cleaning and validation process, we identified and removed some problematic or unsuitable items. As a result, we ended up with 2,864 MCQs, which was not an arbitrary decision. We hope this clarifies our rationale.
>
> Additionally, since the A-Bench dataset is fully manually annotated and requires validation by at least three other humans, **the annotation process is both costly and time-consuming**. As such, it is quite challenging to scale up. We will acknowledge this limitation in our discussion section. However, we also plan to update and expand the dataset in future work, and we appreciate your understanding on this matter.

---

> > ### Author Response · Authors · 2024-11-17
> > **Official Response to Reviewer WKMw**
> >
> > **3. Details of the Image Sampling**
> >
> > **3-A) AIGI sampling for A-Bench-P1.**
> > As mentioned above, **A-Bench-P1** is designed to address the text-alignment issue, so we adopt a manual approach to collect prompts. Specifically, we carefully craft prompts to target key aspects, such as:
> >
> > `Basic Recognition -> Major Object Recognition`: An elaborate treehouse in a thick forest, with children playing inside, rope bridges connecting to other trees, and birds chirping around.
> >
> > `Basic Recognition -> Minor Object Recognition`: A magical fairy ring in a moonlit forest, with tiny glowing fairies dancing and mystical plants all around.
> >
> > `Bag-of-Words -> Attributes Awareness`: A delicate, frosty, crystal snowflake beside a warm, glowing, amber ember on a smooth, slate-gray stone.
> >
> > `Bag-of-Words -> Nouns as Adjectives Awareness`: Shark-sleek submarine exploring ocean depths.
> >
> > `Bag-of-Words -> Composition Identification`: A gamer's setup with consoles and controllers on a desk, multiple screens above, and game boxes and snacks partially obscured beneath the desk.
> >
> > `Bag-of-Words -> Number of Objects Counting`: Six logs in a woodpile, stacked so tightly that they seem to form a solid block.
> >
> > `Outside Knowledge -> Specific Terms Recognition`: A barometer showing a rapid decrease in pressure.
> >
> > `Outside Knowledge -> Contradiction Overcome`: A ship floating above the clouds, sails made of sunlight.
> >
> > To prove content and context diversity, we calculated the **Text Information Entropy** of the text, which resulted in a score of 5.1, indicating high diversity [17]. Additionally, to ensure that the AIGIs cover a broad range of applications, we utilized 15 different AIGI models to generate the images, randomly sampling one AIGI per text prompt. Sample overview can be seen in Fig.6 of the manuscript.
> >
> > **3-B) AIGI sampling for A-Bench-P2**. A-Bench-P2 is designed for the quality evaluation of AIGIs. Consequently, it is essential to ensure that the collected AIGIs span a wide quality range to address various practical scenarios. For `Technical Quality`, we sample 500 AIGIs from the AIGIQA-20K dataset [18] using a uniform sampling strategy. Specifically, each AIGI in the AIGIQA-20K dataset is assigned a mean opinion score (MOS) for technical quality. We apply **uniform sampling** to create more even distributions, as illustrated in Fig. 7 (in the manuscript). For `Aesthetic Quality`, in the absence of provided aesthetic scores, we utilize q-align [19], an effective quality predictor, to infer the aesthetic values of AIGIs. Subsequently, we perform **uniform sampling** similarly to obtain 500 AIGIs for aesthetic evaluation. For `Generative Distortion`, we manually select 500 AIGIs exhibiting unexpected AIGI-specific distortions. It is important to note that there is no content overlap among the selected AIGIs, which can be overviewed in Fig. 8 (in the manuscript).

---

> > > ### Author Response · Authors · 2024-11-17
> > > **Official Response to Reviewer WKMw**
> > >
> > > **References**
> > >
> > > [1] Liu Y, Duan H, Zhang Y, et al. Mmbench: Is your multi-modal model an all-around player?[C]//European Conference on Computer Vision. Springer, Cham, 2025: 216-233.
> > >
> > > [2] Lin Z, Pathak D, Li B, et al. Evaluating text-to-visual generation with image-to-text generation[C]//European Conference on Computer Vision. Springer, Cham, 2025: 366-384.
> > >
> > > [3] Cho J, Hu Y, Garg R, et al. Davidsonian scene graph: Improving reliability in fine-grained evaluation for text-image generation[J]. arXiv preprint arXiv:2310.18235, 2023.
> > >
> > > [4] Ku M, Jiang D, Wei C, et al. Viescore: Towards explainable metrics for conditional image synthesis evaluation[J]. arXiv preprint arXiv:2312.14867, 2023.
> > >
> > > [5] Nichol A, Dhariwal P, Ramesh A, et al. Glide: Towards photorealistic image generation and editing with text-guided diffusion models[J]. arXiv preprint arXiv:2112.10741, 2021.
> > >
> > > [6] Saharia C, Chan W, Saxena S, et al. Photorealistic text-to-image diffusion models with deep language understanding[J]. Advances in neural information processing systems, 2022, 35: 36479-36494.
> > >
> > > [7] Xu P, Shao W, Zhang K, et al. Lvlm-ehub: A comprehensive evaluation benchmark for large vision-language models[J]. arXiv preprint arXiv:2306.09265, 2023.
> > >
> > > [8] Chatterjee A, Stan G B M, Aflalo E, et al. Getting it right: Improving spatial consistency in text-to-image models[C]//European Conference on Computer Vision. Springer, Cham, 2025: 204-222.
> > >
> > > [9] Motamed S, Paudel D P, Van Gool L. Lego: Learning to Disentangle and Invert Personalized Concepts Beyond Object Appearance in Text-to-Image Diffusion Models[J].
> > >
> > > [10] Wang, Y., Zhang, L., Chen, T., & et al.. (2024). Scene graph disentanglement and composition for generalizable complex image generation. arXiv preprint arXiv:2410.00447.
> > >
> > > [11] Huang, L., Zhang, Y., Yang, W., & et al.. (2024). IterComp: Iterative composition-aware feedback learning from model gallery for text-to-image generation. arXiv preprint arXiv:2410.07171.
> > >
> > > [12] Litalby, I., Boulanger, S., & others. (2024). Make It Count: Text-to-Image Generation with an Accurate Number of Objects. arXiv Preprint, 2406.03070.
> > >
> > > [13] Zhou, Y., Xu, W., & Li, X. (2023). Object Count Generation in Diffusion Models. IEEE Transactions on Neural Networks and Learning Systems, 34(11), 2463-2475.
> > >
> > > [14] Schwenk D, Khandelwal A, Clark C, et al. A-okvqa: A benchmark for visual question answering using world knowledge[C]//European conference on computer vision. Cham: Springer Nature Switzerland, 2022: 146-162.
> > >
> > > [15] Vu, H., et al. (2023). WikiContradict: A Benchmark for Evaluating LLMs on Real-World Knowledge Conflicts from Wikipedia. arXiv preprint arXiv:2406.13805.
> > >
> > > [16] Wu H, Zhang Z, Zhang E, et al. Q-bench: A benchmark for general-purpose foundation models on low-level vision[J]. arXiv preprint arXiv:2309.14181, 2023.
> > >
> > > [17] Thomas M, Joy A T. Elements of information theory[M]. Wiley-Interscience, 2006.
> > >
> > > [18] Li C, Kou T, Gao Y, et al. Aigiqa-20k: A large database for ai-generated image quality assessment[J]. arXiv preprint arXiv:2404.03407, 2024, 2(3): 5.
> > >
> > > [19] Wu H, Zhang Z, Zhang W, et al. Q-align: Teaching lmms for visual scoring via discrete text-defined levels[J]. arXiv preprint arXiv:2312.17090, 2023.

---

> ### Author Response · Authors · 2024-11-26
> **Looking forward to discussion.**
>
> Dear Reviewer,
>
> Thank you for recognizing the value of our paper and for providing valuable and constructive feedback. We have carefully addressed the concerns and incorporated additional details based on your thoughtful suggestions in the questions section. We kindly hope that these revisions may merit your consideration for a raised rating.
>
> Should there be any remaining issues or concerns, we would greatly appreciate it if you could kindly point them out, allowing us the opportunity to further discuss and address them.
>
> Thank you once again for your time and thoughtful review.
>
> Best regards,
> The A-Bench Authors

---

### Official Review · Reviewer_8kKW · 2024-11-08

**Soundness:** 2
**Presentation:** 2
**Contribution:** 1
**Rating:** 3
**Confidence:** 5

**Summary:**

Due to the existing evaluation models' inability to effectively assess the performance of AIGI tasks, more and more researchers are turning to LMMs for evaluating the quality of generated images. The authors question this approach and design a framework consisting of seven dimensions focused on high-level semantic understanding and low-level quality evaluation to assess the quality of AIGI. By manually annotating 2864 different image quality issues, the authors compare the evaluation performance of multiple open-source and closed-source LMMs and contrast these with human evaluation results, summarizing numerous shortcomings of LMMs in the AIGI quality assessment task.

**Strengths:**

1. The authors manually annotated a dataset containing 2864 image quality issues, which contributes to the development of AIGI evaluation.
2. The authors evaluate AIGI quality from high-level semantic aspects like counting and low-level aspects like distortion, providing valuable insights for subsequent general AIGI task evaluations.
3. The paper's A-Bench includes the evaluation performance of multiple LMMs, offering guidance for researchers who wish to use LMMs for AIGI quality assessment.

**Weaknesses:**

1. Although A-Bench includes multiple LMMs, it lacks some of the latest SOTA models. Better models such as QWEN-VL2 and MiniCPMv2.6 can be found from opencompass. The paper does not specify the versions of gpt4o used, such as gpt-4o-2024-08-06 or gpt-4o-2024-05-13, which is crucial for future researchers.
2. The AIGI models used to generate the dataset are somewhat outdated, lacking relatively advanced image generation models such as SD3, PixArt, Flux, etc. Currently, the more outstanding AIGI models often embed large language models, which might significantly impact the evaluation conclusions.
3. The questions are all manually generated, which is certainly good. However, this makes the evaluation dataset difficult to expand and might lose value as AIGI models rapidly evolve. It would be better if the questions could be designed based on the text prompts of T2I models.
4. Compared to previous work, The paper's main contribution, i.e., high-level semantic question answering, is not strongly related to AIGI and does not seem necessary to research specifically in the AIGI context.
5. Two-thirds of the low-level semantic question-answering data come from other datasets, reducing the paper's contribution.
6. The paper's findings are somewhat unremarkable. It is obvious that closed-source LMMs perform better than open-source ones, and some other findings, such as the LMMs' insufficient perception of distortion, have already been mentioned in works like Q-Bench.

**Questions:**

1. Overall, the paper adds the evaluation of LMMs' high-level semantic cognition for AIGI, in addition to previous work using LMMs to assess image generation quality. However, it does not highlight the difference between AIGI tasks and conventional cognition tasks. Could the authors elaborate on this further?
2. Generally, the authors focus more on evaluating the perceptual capabilities of LMMs, but these perceptual capabilities are more inclined towards the low-level aspects for AIGI tasks. Could the authors further elaborate on the differences between the low-level perceptual aspects of AIGI and some earlier works?

---

> ### Author Response · Authors · 2024-11-17
> **Official Responses to Reviewer 8kKW**
>
> We would like to thank the reviewer for the time and meaningful comments. First, we would like to kindly clarify a critical point that weakness 5 "Two-thirds of the low-level semantic question-answering data come from other datasets" is a misunderstanding. **All the Question-Answering data in A-Bench is entirely original and created by us and none of it comes from any other existing datasets.** We will then address your concerns point-by-point.
>
> **1. Adding Latest SOTA models**
>
> Thank you for your suggestions. We have further tested five additional models. Please note that **part of the correct answers in the A-Bench dataset are kept private** (to avoid data leak). The updated performance is shown in the table. Additionally, the version of GPT-4o used in the paper is GPT-4o-2024-05-13. For completeness, we have also included GPT-4O-2024-08-06 for comparison.
>
> | LMM             | Basic Recognition | Bag-of-Words | Outside Knowledge | Technical | Aesthetic | Generative | Overall |
> |-----------------|-------------------|--------------|-------------------|-----------|-----------|------------|---------|
> | GPT-4o （2024-08-06）          | 0.939           | 0.832     | 0.678     | 0.703      | 0.621      | 0.676       | 0.758   |
> | GPT-4o （2024-05-13）           | 0.947           | 0.813     | 0.675     | 0.706      | 0.616      | 0.679       | 0.759   |
> | **Qwen2-VL-72B**                      | 0.949         | 0.822       | 0.701      | 0.742      | 0.603      | 0.702      | **0.767**    |
> | MiniCPM-V-2.6                        | 0.934              | 0.910         | 0.699              | 0.691      | 0.601      | 0.605       | 0.744    |
> | InternVL2-40B                         | 0.947             | 0.920         | 0.697              | 0.663     | 0.632      | 0.501       | 0.752    |
> | Ovis1.5-Llama3-8B                 | 0.931              | 0.924         | 0.692              | 0.708      | 0.678      | 0.554       | 0.751    |
> | LLaVA-OneVision-7B              | 0.929              | 0.924         | 0.695              | 0.688      | 0.678      | 0.543       | 0.748    |
>
> **2. Timeliness of A-Bench**
>
> Creating a benchmark involves generating images, collecting data, training evaluators, and verifying data quality, making the process both time-consuming and costly. As a result, it is inevitable that AIGI benchmarks may not always keep pace with the latest technologies or models. However, the insights provided by the benchmark in evaluating AIGI remain valuable and offer useful guidance. We will address this limitation in the discussion section. Of course, we are committed to ongoing updates and expansions to ensure the benchmark remains current. We appreciate your understanding.
>
> **3. Question Annotation**
>
> Thank you for your comment. **While human annotations require significant time and may cause AIGI benchmarks to lag behind the rapid evolution of technology, they remain essential.** Human annotations are critical for AIGI benchmarks because **they provide accurate, consistent, and context-sensitive evaluations that AI models alone cannot achieve**, particularly when assessing subjective qualities such as creativity, coherence, and alignment with complex prompts. Although human involvement makes it more challenging to scale datasets and can delay the integration of the latest model advancements, it ensures reliable ground truth, addresses edge cases, and upholds ethical and cultural sensitivity. Moreover, human reviewers are capable of evaluating complex, nuanced aspects of generated images that current AI models cannot effectively assess, helping to ensure the benchmark remains both meaningful and trustworthy, even as AIGI technologies rapidly evolve.
>
> Thank you for your suggestion regarding using prompts for annotation, we find it very valuable. However, there are exceptions, such as cases involving prompts that require outside knowledge, where the AIGI may not generate outputs as expected, necessitating human verification. Additionally, for visual quality annotations, the prompt itself has limited relevance. We also recognize that prompts could constrain annotators' creativity, potentially overlooking interesting and diverse questions. Nonetheless, we will consider your suggestion in future annotation efforts. Thank you again for your insightful feedback.

---

> > ### Author Response · Authors · 2024-11-17
> > **Official Responses to Reviewer 8kKW**
> >
> > **4. Relation between High-level Semantic Question Answering and AIGI**
> >
> > Thanks for your question. We would like to emphasize that **our proposed High-level Semantic Question Answering is strongly linked to AIGI evaluation, particularly in the area of alignment evaluation.**
> >
> >
> > When evaluating the semantic capabilities of LMMs, it is generally recommended to test from simpler to more complex tasks [6]. Thus, our aspect selection **follows a progression from basic to more complex dimensions**. Since our primary focus is on exploring LMM-based AIGI evaluations, we draw on previous work that uses LMMs for evaluation metric design [1, 2, 3], where LMMs are typically employed for image-related question-answer tasks.
> > To evaluate the alignment between an AIGI and its prompt, we first assess overall subject alignment, which led us to select the first dimension, `Basic Recognition`. We then evaluate the attributes, actions, and interactions of objects in the AIGI, leading to the second dimension, `Bag-of-Words Pitfalls Discrimination`. Finally, given the creative nature of AIGI generation, which often requires external knowledge, we chose the third dimension, `Outside Knowledge Realization`.
> >
> > We further refined our dimensions by considering specific aspects of AIGI generation that are particularly relevant:
> >
> > 1. `Major Object Recognition` and `Minor Object Recognition` focus on identifying generated objects, which is a fundamental capability of AIGIs [4, 5].
> > 2. `Attributes Awareness` evaluates the model's sensitivity to object attributes, which is crucial for basic evaluations [6, 7].
> > 3.  `Nouns as Adjectives Awareness` addresses potential issues where T2I models may misinterpret nouns as adjectives, generating objects instead of intended attributes [8, 9].
> > 4. `Composition Identification` pertains to understanding compositional relationships [10, 11].
> > 5. `Number of Objects Counting` assesses the model's ability to accurately count objects, which is critical for checking if the AIGI matches numerical specifications in the prompt [12, 13].
> > 6. `Specific Terms Recognition` involves identifying domain-specific scenes and objects, such as geography, sports, or food, important for external knowledge [14].
> > 7. `Contradiction Overcome` tests the model's ability to correctly interpret AIGIs even when their content contradicts established world knowledge [15].
> >
> > Thus, our dimension selection follows **a suggested benchmark approach from simple to complex, with sub-dimensions designed to address critical points in AIGI generation and LMM evaluation.** Therefore, the High-level Semantic Question Answering framework we propose is indeed tailored to support LMM evaluation on AIGIs.
> >
> > **5. Misunderstanding of the Question-Answering Data**
> >
> > The reviewer might have a misunderstanding here. We would like to clarify that **all the Question-Answering data in A-Bench is entirely original and is created by us, it does not come from any other existing datasets**. Specifically, each Question-answer pair in A-Bench is first annotated by a trained annotator and then verified by three additional reviewers before being finalized.
> >
> > The only data sourced from an external dataset is the A-Bench-P2 images, which are sampled from AIGIQA-20K [16]. However, we would like to reiterate that **all the Question-Answering data in A-Bench is collected and created by us.** We hope our clarification helps clear up any misunderstanding.

---

> > > ### Author Response · Authors · 2024-11-17
> > > **Official Responses to Reviewer 8kKW**
> > >
> > > **6. Meaning of the findings**
> > >
> > > Thank you for your concern. **While it may seem obvious that closed-source LMMs generally outperform open-source ones, the specific gaps and their underlying causes are worth exploring.** For instance, the best-performing open-source model in `Basic Recognition` is actually quite close to the closed-source models. However, the difference becomes more pronounced in tasks like `Composition Identification` and `Number of Objects Counting`, highlighting areas where open-source models still have room for improvement. If these open-source models are used as evaluators for AIGI, their lower performance in these dimensions could explain why they underperform compared to closed-source models. **Therefore, while the general finding may appear obvious, the detailed comparison in A-Bench offers valuable insights into the specific strengths and weaknesses of these models.** We will also emphasize these points in the discussion section to highlight our contributions.
> > >
> > > Regarding your point that "LMMs' insufficient perception of distortion has already been mentioned in works like Q-Bench," we would like to clarify that **this is only a part of our conclusion in A-Bench-P2**. We further discuss **how most LMMs exhibit their weakest performance** in the `Generative Distortion Assessment` subcategory. Additionally, we highlight an interesting finding: while humans typically perform better in` Technical Quality Perception` compared to `Aesthetic Quality Evaluation`, LMMs show similar performance levels in both subcategories—an analysis not covered in Q-Bench. Therefore, relying solely on this familiar conclusion to dismiss the more detailed contributions in our later discussion is not entirely appropriate. We hope for your understanding.
> > >
> > > **7. Difference Between AIGI tasks and Conventional Cognition Tasks**
> > >
> > > Thank you for your question. **The key distinction is that traditional cognition tasks are designed to assess a model's understanding capabilities, whereas A-Bench focuses on using cognition tasks to diagnose potential issues in LMM evaluation, specifically related to AIGI generation**. As mentioned in our response to the fourth point, our proposed **High-level Semantic Question Answering is closely tied to AIGI evaluation, particularly in alignment assessment.** By evaluating LMM performance across different semantic dimensions in AIGI, we aim to identify evaluation challenges and suggest possible areas for improvement.
> > >
> > > **8. Differences Between the Low-level Perceptual Aspects of AIGI and Some Earlier Works**
> > >
> > > Thank you for your question. The key difference between the **low-level perceptual aspects** in A-Bench and previous works such as Q-Bench [17] and DepictQA [18] lies in the focus and optimization for AIGI tasks. Previous works primarily focus on **general perceptual dimensions in traditional Image Quality Assessment (IQA), often targeting natural images, and do not specifically optimize for AIGI low-level evaluation.** For instance, Q-Bench broadly categorizes low-level perceptual aspects into `distortions` and `other attributes`, without addressing AIGI-specific issues.
> > >
> > > In contrast, **A-Bench systematically decouples low-level perceptual evaluation into three distinct areas: `technical`, `aesthetic`, and `generative distortion`.** The design of `technical` and `aesthetic` dimensions is based on the fact that AIGIs share certain common low-level aspects with traditional IQA in these areas. However, in particular, A-Bench **places a strong emphasis on generative distortion**, which includes issues like generative blur (typically caused by incomplete generation, distinct from traditional motion blur or compression artifacts), confusing geometric structures, and unnaturalness. We have specifically designed question-answer pairs to assess these generative distortions. This focus marks the most significant distinction between A-Bench and earlier works in terms of low-level perceptual evaluation.

---

> > > > ### Author Response · Authors · 2024-11-17
> > > > **Official Responses to Reviewer 8kKW**
> > > >
> > > > **References**
> > > >
> > > > [1] Lin Z, Pathak D, Li B, et al. Evaluating text-to-visual generation with image-to-text generation[C]//European Conference on Computer Vision. Springer, Cham, 2025: 366-384.
> > > >
> > > > [2] Cho J, Hu Y, Garg R, et al. Davidsonian scene graph: Improving reliability in fine-grained evaluation for text-image generation[J]. arXiv preprint arXiv:2310.18235, 2023.
> > > >
> > > > [3] Ku M, Jiang D, Wei C, et al. Viescore: Towards explainable metrics for conditional image synthesis evaluation[J]. arXiv preprint arXiv:2312.14867, 2023.
> > > >
> > > > [4] Nichol A, Dhariwal P, Ramesh A, et al. Glide: Towards photorealistic image generation and editing with text-guided diffusion models[J]. arXiv preprint arXiv:2112.10741, 2021.
> > > >
> > > > [5] Saharia C, Chan W, Saxena S, et al. Photorealistic text-to-image diffusion models with deep language understanding[J]. Advances in neural information processing systems, 2022, 35: 36479-36494.
> > > >
> > > > [6] Liu Y, Duan H, Zhang Y, et al. Mmbench: Is your multi-modal model an all-around player?[C]//European Conference on Computer Vision. Springer, Cham, 2025: 216-233.
> > > >
> > > > [7] Xu P, Shao W, Zhang K, et al. Lvlm-ehub: A comprehensive evaluation benchmark for large vision-language models[J]. arXiv preprint arXiv:2306.09265, 2023.
> > > >
> > > > [8] Chatterjee A, Stan G B M, Aflalo E, et al. Getting it right: Improving spatial consistency in text-to-image models[C]//European Conference on Computer Vision. Springer, Cham, 2025: 204-222.
> > > >
> > > > [9] Motamed S, Paudel D P, Van Gool L. Lego: Learning to Disentangle and Invert Personalized Concepts Beyond Object Appearance in Text-to-Image Diffusion Models[J].
> > > >
> > > > [10] Wang, Y., Zhang, L., Chen, T., & et al.. (2024). Scene graph disentanglement and composition for generalizable complex image generation. arXiv preprint arXiv:2410.00447.
> > > >
> > > > [11] Huang, L., Zhang, Y., Yang, W., & et al.. (2024). IterComp: Iterative composition-aware feedback learning from model gallery for text-to-image generation. arXiv preprint arXiv:2410.07171.
> > > >
> > > > [12] Litalby, I., Boulanger, S., & others. (2024). Make It Count: Text-to-Image Generation with an Accurate Number of Objects. arXiv Preprint, 2406.03070.
> > > >
> > > > [13] Zhou, Y., Xu, W., & Li, X. (2023). Object Count Generation in Diffusion Models. IEEE Transactions on Neural Networks and Learning Systems, 34(11), 2463-2475.
> > > >
> > > > [14] Schwenk D, Khandelwal A, Clark C, et al. A-okvqa: A benchmark for visual question answering using world knowledge[C]//European conference on computer vision. Cham: Springer Nature Switzerland, 2022: 146-162.
> > > >
> > > > [15] Vu, H., et al. (2023). WikiContradict: A Benchmark for Evaluating LLMs on Real-World Knowledge Conflicts from Wikipedia. arXiv preprint arXiv:2406.13805.
> > > >
> > > > [16] Li C, Kou T, Gao Y, et al. Aigiqa-20k: A large database for ai-generated image quality assessment[J]. arXiv preprint arXiv:2404.03407, 2024, 2(3): 5.
> > > >
> > > > [17] Wu H, Zhang Z, Zhang E, et al. Q-bench: A benchmark for general-purpose foundation models on low-level vision[J]. arXiv preprint arXiv:2309.14181, 2023.
> > > >
> > > > [18] You Z, Gu J, Li Z, et al. Descriptive image quality assessment in the wild[J]. arXiv preprint arXiv:2405.18842, 2024.

---

> ### Author Response · Authors · 2024-11-26
> **Looking forward to discussion.**
>
> Dear Reviewer,
>
> Thank you for your valuable and constructive feedback. We have carefully addressed your comments and revised the paper accordingly. We kindly ask if these revisions have resolved your concerns. If so, we would greatly appreciate your consideration of a raised rating.
>
> If there are any remaining issues or concerns, we would be grateful if you could kindly point them out, allowing us the opportunity to discuss and address them further.
>
> Thank you for your time and thoughtful review.
>
> Best regards,
> A-Bench authors

---

### Meta-Review · Area_Chair_QHpv · 2024-12-19

**Metareview:**

Summary
The paper examines whether multimodal models can evaluate image generation models. The authors propose a benchmark, A-Bench, that can evaluate both the text alignment and the image quality of generation models. A-Bench is a diagnostic benchmark, and the authors use different multimodal models. Their key finding is that not all multimodal models can serve as evaluators, and certain close-sourced models are more suited for this task.

Strengths
1. The paper studies an important problem: easing the evaluation of generation models by using image understanding models.
2. The experiments in this paper cover a wide variety of models for both generation and understanding. This comprehensive study is valuable.
3. The conclusions drawn in the paper are novel and important for the research community.

Weaknesses
1. The size of the diagnostic benchmark seems small. While the authors do provide the comparison to MMBench, I believe MMBench is used in conjunction with multiple benchmarks. If the authors intend such a use for A-Bench, it would be good to clarify in the paper.
2. Unexplained abbreviations as also pointed out by one reviewer.

Suggestions
Maybe using a stronger Llama image understanding model can strengthen the paper ? The closed sourced systems on GPT and Gemini are hard to use at scale for many researchers.

Justification
This is a well written paper that studies an important problem. Technically sound and well executed.

**Additional Comments On Reviewer Discussion:**

The authors engaged with the reviewers to address questions. 2 of the reviewers didn't engage with the authors, but the AC has read through the reviews and believes that their concerns were already answered.

---

### Decision · Program_Chairs · 2025-01-22

Accept (Poster)